# Proximity Operator of the Matrix Perspective Function and its Applications

**Joong-Ho Won**
Department of Statistics
Seoul National University
wonj@stats.snu.ac.kr

## Abstract

We show that the matrix perspective function, which is jointly convex in the Cartesian product of a standard Euclidean vector space and a conformal space of symmetric matrices, has a proximity operator in an almost closed form. The only implicit part is to solve a semismooth, univariate root finding problem. We uncover the connection between our problem of study and the matrix nearness problem. Through this connection, we propose a quadratically convergent Newton algorithm for the root finding problem. Experiments verify that the evaluation of the proximity operator requires at most 8 Newton steps, taking less than 5s for 2000 by 2000 matrices on a standard laptop. Using this routine as a building block, we demonstrate the usefulness of the studied proximity operator in constrained maximum likelihood estimation of Gaussian mean and covariance, peudolikelihood-based graphical model selection, and a matrix variant of the scaled lasso problem.

## 1 Introduction

The main theme of this paper is the proximity operator of the matrix perspective function, defined as

$$\phi(\boldsymbol{\Omega}, \boldsymbol{\eta}) = \begin{cases} \frac{1}{2}\boldsymbol{\eta}^T\boldsymbol{\Omega}^\dagger\boldsymbol{\eta}, & \boldsymbol{\Omega} \succeq \boldsymbol{0}, \ \boldsymbol{\eta} \in \mathcal{R}(\boldsymbol{\Omega}), \\ \infty, & \text{otherwise}, \end{cases}$$

for $\boldsymbol{\eta} \in \mathbb{R}^p$, the $p$-dimensional Euclidean space, and $\boldsymbol{\Omega} \in \mathbb{S}^p$, the vector space of $p \times p$ symmetric matrices. Matrix $\boldsymbol{\Omega}^\dagger$ is the Moore-Penrose pseudoinverse of $\boldsymbol{\Omega}$. The range of $\boldsymbol{\Omega}$ is denoted by $\mathcal{R}(\boldsymbol{\Omega})$. Relation $\succeq$ refers to the Löwner partial order of matrices, i.e., $\boldsymbol{A} \succeq \boldsymbol{B}$ means that $\boldsymbol{A} - \boldsymbol{B}$ is positive semidefinite. Function $\phi$ is jointly convex in $\boldsymbol{\Omega}$ and $\boldsymbol{\eta}$. An easy way to see this is to note that

$$\phi(\boldsymbol{\Omega}, \boldsymbol{\eta}) = \sup_{\boldsymbol{w} \in \mathbb{R}^p} \left[ \boldsymbol{\eta}^T\boldsymbol{w} - \frac{1}{2}\boldsymbol{w}^T\boldsymbol{\Omega}\boldsymbol{w} \right] \tag{1}$$

[22, p. 70]. The supremand is linear in $(\boldsymbol{\Omega}, \boldsymbol{\eta})$; a supremum of linear functions is convex. We will shortly see that $\phi$ is also closed (lower semicontinuous). The name "matrix perspective" comes from the perspective of a function frequently encountered in convex analysis. The (closed) perspective $g : \mathbb{R}^d \times \mathbb{R} \to \mathbb{R} \cup \{\infty\}$ of a closed convex function $f : \mathbb{R}^d \to \mathbb{R} \cup \{\infty\}$ is defined as the closure of function $\tilde{g}(t, \boldsymbol{x}) = tf(t^{-1}\boldsymbol{x})$ if $t > 0$, and $\infty$ otherwise [9, 18].

The proximity operator of a convex function $f : \mathbb{R}^d \to \mathbb{R} \cup \{+\infty\}$ is uniquely defined and denoted

$$\mathbf{prox}_{\gamma f}(\boldsymbol{x}) = \underset{\boldsymbol{u} \in \mathbb{R}^d}{\arg\min} \left[ f(\boldsymbol{u}) + \frac{1}{2\gamma}\|\boldsymbol{u} - \boldsymbol{x}\|_2^2 \right], \quad \gamma > 0.$$

If we restrict $\boldsymbol{\Omega}$ to be $\boldsymbol{\Omega} = t\boldsymbol{I}_p$ where $\boldsymbol{I}_p$ is the identity matrix of order $p$, then $\bar{\phi}(t, \boldsymbol{\eta}) := \phi(t\boldsymbol{I}_p, \boldsymbol{\eta})$ becomes the conventional perspective of the squared Euclidean norm function $\frac{1}{2}\|\cdot\|_2^2$. In this special case, a closed-form representation of the proximity operator of $\bar{\phi}$ has recently been found [10].

## 1.1 Applications of the matrix perspective function and its proximity operator

The motivation for studying the function $\phi$ is its ubiquity in machine learning and statistics. We provide three examples:

**Gaussian joint likelihood estimation** It is well-known that the negative log-likelihood of a $p$-variate Gaussian mean-covariance pair $(\boldsymbol{\mu}, \boldsymbol{\Sigma})$ given data $\{\boldsymbol{x}_1, \ldots, \boldsymbol{x}_N\}$ is

$$\tilde{\ell}(\boldsymbol{\Sigma}, \boldsymbol{\mu}) = \log \det \boldsymbol{\Sigma} + \mathbf{Tr}(\boldsymbol{\Sigma}^{-1} \boldsymbol{S}) - 2\bar{\boldsymbol{\mu}}^T \boldsymbol{\Sigma}^{-1} \boldsymbol{\mu} + \boldsymbol{\mu}^T \boldsymbol{\Sigma}^{-1} \boldsymbol{\mu},$$

up to scaling and additive constant, where $\bar{\boldsymbol{\mu}} = \frac{1}{N} \sum_{i=1}^{N} \boldsymbol{x}_i$ and $\boldsymbol{S} = \frac{1}{N} \sum_{i=1}^{N} \boldsymbol{x}_i \boldsymbol{x}_i^T$; $\mathbf{Tr}(\boldsymbol{M})$ is the trace of matrix $\boldsymbol{M}$. If we change the variables to $\boldsymbol{\Omega} = \boldsymbol{\Sigma}^{-1}$ and $\boldsymbol{\eta} = \boldsymbol{\Omega} \boldsymbol{\mu}$, then

$$\tilde{\ell}(\boldsymbol{\Sigma}, \boldsymbol{\mu}) = \ell(\boldsymbol{\Omega}, \boldsymbol{\eta}) = -\log \det \boldsymbol{\Omega} + \mathbf{Tr}(\boldsymbol{\Omega} \boldsymbol{S}) - 2\bar{\boldsymbol{\mu}}^T \boldsymbol{\eta} + \phi(\boldsymbol{\Omega}, \boldsymbol{\eta}). \tag{2}$$

Function $\ell$ is not coercive unless $\boldsymbol{S}$ is positive definite. Constraints encoding the prior knowledge can be added to ensure existence (and/or uniqueness) of the solution. An example is upper bounds on the variances: if $\boldsymbol{y} \sim \mathcal{N}(\mu, \boldsymbol{\Sigma})$, then $\mathbf{var}\,[\boldsymbol{c}_i^T \boldsymbol{y}] \leq 1$ translates to $\boldsymbol{c}_i^T \boldsymbol{\Omega}^{-1} \boldsymbol{c}_i = 2\phi(\boldsymbol{\Omega}, \boldsymbol{c}_i) \leq 1$, which is convex for given $\boldsymbol{c}_i \in \mathbb{R}^p$, $i = 1, \ldots, m$.

**Graphical model selection** In Gaussian graphical models, the pseudolikelihoood [1] of the precision matrix $\boldsymbol{\Omega}$ given data matrix $\boldsymbol{Y} = [\boldsymbol{y}_1, \ldots, \boldsymbol{y}_N]^T$ is

$$PL(\boldsymbol{\Omega}) = \frac{N}{2} \sum_{i=1}^{p} \log \omega_{ii} - \frac{1}{2} \sum_{i=1}^{N} \sum_{j=1}^{p} \omega_{ii}^{-1} (\sum_{k=1}^{p} \omega_{ik} y_{jk})^2 = \frac{N}{2} \log \det \boldsymbol{\Omega}_D - N\phi(\mathcal{K}\boldsymbol{\Omega}) \tag{3}$$

for $\boldsymbol{\Omega} = (\omega_{ij}) = \boldsymbol{\Sigma}^{-1}$ and $\boldsymbol{\Omega}_D = \mathrm{diag}(\omega_{11}, \ldots, \omega_{pp})$; $\mathcal{K} : \boldsymbol{\Omega} \mapsto \frac{1}{N}(\boldsymbol{I}_N \otimes \boldsymbol{\Omega}_D, \mathbf{vec}(\boldsymbol{\Omega}\boldsymbol{Y}^T))$ is a linear map, where $\otimes$ is the Kronecker product and $\mathbf{vec}$ is the usual vectorization operator. Often a sparsity-inducing penalty $-\lambda \sum_{i<j} |\omega_{ij}|$ is added to the pseudolikelihood and the sum is maximized.

**Heteroskedastic scaled lasso** The scaled lasso [32] minimizes

$$\ell(\sigma, \boldsymbol{\beta}) = \frac{1}{2\sigma} \|\boldsymbol{y} - \boldsymbol{X}\boldsymbol{\beta}\|_2^2 + \frac{\sigma}{2} + \lambda\|\boldsymbol{\beta}\|_1$$

for the linear model $\boldsymbol{y} = \boldsymbol{X}\boldsymbol{\beta} + \boldsymbol{\epsilon}$, where $\boldsymbol{X} \in \mathbb{R}^{N \times p}$ is the data matrix, and $\boldsymbol{\epsilon} \sim \mathcal{N}(0, \sigma^2 \boldsymbol{I}_N)$.

This estimation problem can be extended to a heteroskedastic setting, i.e., $\boldsymbol{\epsilon} \sim \mathcal{N}(0, \boldsymbol{\Sigma})$: for $\boldsymbol{\Omega} = \boldsymbol{\Sigma}^{1/2}$, we minimize

$$\ell(\boldsymbol{\Omega}, \boldsymbol{\beta}) = \phi(\boldsymbol{\Omega}, \boldsymbol{X}\boldsymbol{\beta} - \boldsymbol{y}) + \frac{1}{2\sqrt{N}}\|\boldsymbol{\Omega}\|_F + \lambda\|\boldsymbol{\beta}\|_1. \tag{4}$$

where $\|\boldsymbol{M}\|_F = [\mathbf{Tr}(\boldsymbol{M}^T\boldsymbol{M})]^{1/2}$ is the Frobenius norm of matrix $\boldsymbol{M}$.

**Proximal algorithms** All of these examples distill to the convex optimization problem:

$$\min_{\boldsymbol{\Omega} \in \mathbb{S}^p, \boldsymbol{\eta} \in \mathbb{R}^p} f(\boldsymbol{\Omega}, \boldsymbol{\eta}) + g(\boldsymbol{\Omega}, \boldsymbol{\eta}) + h(\mathcal{K}[\boldsymbol{\Omega}, \boldsymbol{\eta}]), \tag{5}$$

where $f$, $g$, and $h$ are convex with $f$ differentiable, and $\mathcal{K}$ is an affine map. Either $g = \phi$ or $h = \phi$, depending on the problem. Since $\phi$ (and possibly other components of (5)) is nonsmooth, conventional solution methods are difficult to apply, especially when the problem size is large. In this setting, proximal algorithms such as the primal-dual hybrid gradient (PDHG) method [6, 11, 12, 14, 20, 21, 34, 37] can be applied. In particular, following [12, 20, 34], we obtain

$$(\boldsymbol{\Omega}^{k+1}, \boldsymbol{\eta}^{k+1}) = \mathbf{prox}_{\tau g}\left((\boldsymbol{\Omega}^k, \boldsymbol{\eta}^k) - \tau\big(\nabla f(\boldsymbol{\Omega}^k, \boldsymbol{\eta}^k) + \mathcal{K}^T[\boldsymbol{\Theta}^k, \boldsymbol{\zeta}^k]\big)\right)$$

$$(\tilde{\boldsymbol{\Omega}}^{k+1}, \tilde{\boldsymbol{\eta}}^{k+1}) = (2\boldsymbol{\Omega}^{k+1} - \boldsymbol{\Omega}^k, 2\boldsymbol{\eta}^{k+1} - \boldsymbol{\eta}^k) \tag{6}$$

$$(\boldsymbol{\Theta}^{k+1}, \boldsymbol{\zeta}^{k+1}) = \mathbf{prox}_{\sigma h^*}\left((\boldsymbol{\Theta}^k, \boldsymbol{\zeta}^k) + \sigma\mathcal{K}[\tilde{\boldsymbol{\Omega}}^{k+1}, \tilde{\boldsymbol{\eta}}^{k+1}]\right),$$

where $\mathcal{K}^T$ is the adjoint of the linear part of $\mathcal{K}$, and $h^*(\boldsymbol{\Theta}, \boldsymbol{\zeta}) = \sup_{\boldsymbol{\Omega}, \boldsymbol{\eta}} \langle (\boldsymbol{\Omega}, \boldsymbol{\eta}), (\boldsymbol{\Theta}, \boldsymbol{\zeta}) \rangle - h(\boldsymbol{\Omega}, \boldsymbol{\eta})$ is the Fenchel conjugate of $h$. Convergence to a solution to problem (5) occurs if the step sizes $(\sigma, \tau)$ satisfy $\tau(L_f/2 + \sigma\|\mathcal{K}^T\mathcal{K}\|_2) < 1$. Here $L_f$ is the Lipschitz modulus of the gradient of $f$, and $\|\cdot\|_2$ is the operator 2-norm of the linear part of an affine operator. Moreau's decomposition

$$(\boldsymbol{\Omega}, \boldsymbol{\eta}) = \mathbf{prox}_{\sigma h^*}(\boldsymbol{\Omega}, \boldsymbol{\eta}) + \sigma\,\mathbf{prox}_{\sigma^{-1} h}(\sigma^{-1}(\boldsymbol{\Omega}, \boldsymbol{\eta})) \tag{7}$$

confirms the practical importance of $\mathbf{prox}_\phi$. Yet, the latter does not offer a closed form expression. Hence, efficient computation of $\mathbf{prox}_\phi$ is a key to success of solving the above learning problems.

## 1.2 Contributions

The contributions of this paper are 1) to show that evaluation of the proximity operator of $\phi$ reduces to finding the unique root of a univariate function — given the root, the operator takes a closed form; 2) to reveal the unexpected connection between the proximity operator and the matrix nearness problem [16]; 3) to develop a quadratically convergent Newton algorithm for root-finding despite the nonsmoothness of the function, made possible by exploiting the connection; 4) to investigate novel applications of proximal optimization methods in learning problems.

## 2 Characterization of the proximity operator via matrix nearness

In this section we characterize the proximity operator of $\phi$ in terms of the root of a univariate function. This is achieved by showing that the dual of the optimization problem involved with the operator is a matrix nearness problem, which is to find, for an arbitrary matrix, a nearest (in terms of a matrix norm) member of some given class of matrices [2, 4, 16, 17, 24, 28]. To our knowledge, the connection between the matrix perspective function and the matrix nearness problem is first uncovered.

We frequently use the following fact and notation: any symmetric matrix $M$ admits a unique (and explicit) decomposition $M = M_+ - M_-$ such that $M_+, M_- \succeq 0$. If $M$ has a spectral decomposition $Q \operatorname{diag}(\nu_1, \ldots, \nu_n) Q^T$, then $M_+ = Q \operatorname{diag}((\nu_1)_+, \ldots, (\nu_n)_+) Q^T$, where $\nu_+ = \max(0, \nu)$. We also denote $\nu_i$ by $\lambda_i(M)$, and let $n = p + 1$ in the sequel.

Recall the variational formulation (1) of the matrix perspective function $\phi$ is a convex quadratic programming (QP) problem. An equivalent formulation of this QP is

$$
\begin{array}{ll}
\max_{w \in \mathbb{R}^p, V \in \mathbb{S}^p} & \operatorname{Tr}(\Omega V) + \eta^T w \\
\text{subject to} & V = -\frac{1}{2} w w^T
\end{array},
$$

which in turn is equivalent to the following SDP:

$$
\begin{array}{ll}
\max_{w \in \mathbb{R}^p, V \in \mathbb{S}^p} & \operatorname{Tr}(\Omega V) + w^T \eta \\
\text{subject to} & \begin{bmatrix} -V & \frac{1}{\sqrt{2}} w \\ \frac{1}{\sqrt{2}} w^T & 1 \end{bmatrix} \succeq 0
\end{array}
$$

since the relaxation $V + \frac{1}{2} w w^T \preceq 0$ of the equality constraint $V + \frac{1}{2} w w^T = 0$ is tight [3, pp. 653–654]; the Schur complement shows that the above linear matrix inequality constraint is equivalent to this nonconvex relaxation. Define a closed convex cone

$$
C = \left\{ (V, w) \in \mathbb{S}^p \times \mathbb{R}^p : V + \frac{1}{2} w w^T \preceq 0 \right\} \tag{8}
$$

and note that $\operatorname{Tr}(\Omega V) + w^T \eta$ is the standard inner product of the vector space $\mathbb{S}^p \times \mathbb{R}^p$; we can write $\operatorname{Tr}(\Omega V) + w^T \eta = \langle (\Omega, w), (V, \eta) \rangle$. Then we see that

$$
\phi(\Omega, \eta) = \sigma_C(\Omega, \eta),
$$

where $\sigma_S(x) = \sup_{y \in S} \langle x, y \rangle$ is the support function of a set $S$. Elementary convex analysis results tell us that $\sigma_C$ is closed, convex, proper, and the Fenchel conjugate function of the $0/\infty$ indicator function $\iota_C(V, w)$ of $C$. (Hence we have shown that $\phi$ is closed.) From Moreau's decomposition (7), if we denote the projection onto $C$ by $P_C$, then

$$
\mathbf{prox}_{\gamma \phi}(X, y) = (X, y) - \gamma P_C(\gamma^{-1} X, \gamma^{-1} y), \tag{9}
$$

since the proximity operator of $\iota_C$ is $P_C$.

To compute $P_C(X, y)$, we need to solve the SDP

$$
\begin{array}{ll}
\min_{V, w} & \frac{1}{2} \|w - y\|_2^2 + \frac{1}{2} \|V - X\|_F^2 \\
\text{subject to} & \begin{bmatrix} -V & \frac{1}{\sqrt{2}} w \\ \frac{1}{\sqrt{2}} w^T & 1 \end{bmatrix} \succeq 0.
\end{array} \tag{10}
$$

If $(V^\star, w^\star)$ solves problem (10), then $P_C(X, y) = (V^\star, w^\star)$. If $(X, y) \in C$, then clearly $(V^\star, w^\star) = (X, y)$. Thus we focus on the case $(X, y) \notin C$. Construct block matrices

$$
U = \begin{bmatrix} -V & \frac{1}{\sqrt{2}} w \\ \frac{1}{\sqrt{2}} w^T & 1 \end{bmatrix}, \quad \bar{X} = \begin{bmatrix} -X & \frac{1}{\sqrt{2}} y \\ \frac{1}{\sqrt{2}} y^T & 1 \end{bmatrix}. \tag{11}
$$

Set $e = (0, \dots, 0, 1)^T \in \mathbb{R}^n$. Then problem (10) is equivalent to

$$
\begin{aligned}
\min_U \quad & \tfrac{1}{2}\|U - \bar{X}\|_F^2 \\
\text{subject to} \quad & U \succeq 0, \; e^T U e = 1.
\end{aligned}
\tag{12}
$$

This is a special case of the *least-squares covariance matrix adjustment problem* [4], an instance of the matrix nearness problem. Following [4], we minimize the Lagrangian

$$
\mathcal{L}(U, \Lambda, \mu) = \frac{1}{2}\|U - \bar{X}\|_F^2 - \mathrm{Tr}(\Lambda U) + \mu(e^T U e - 1), \quad \Lambda \succeq 0,
$$

with respect to $U$, to obtain the dual objective function:

$$
\tilde{g}(\Lambda, \mu) = -\frac{1}{2}\|\Lambda - \mu e e^T + \bar{X}\|_F^2 + \frac{1}{2}\|\bar{X}\|_F^2 - \mu, \quad \Lambda \succeq 0.
\tag{13}
$$

If $(\Lambda^\star, \mu^\star)$ maximizes function (13), then the solution to primal (12) is recovered by the relation

$$
U^\star = \bar{X} - \mu^\star e e^T + \Lambda^\star,
\tag{14}
$$

since strong duality holds ($U = I$ is strictly feasible).

The dual problem reduces to a univariate convex optimization problem in $\mu$. By partially maximizing the objective (13) over $\Lambda \succeq 0$ with $\mu$ fixed, we see the minimizer is

$$
\Lambda(\mu) = \arg\min_{\Lambda \succeq 0} \frac{1}{2}\left\|\Lambda - (\mu e e^T - \bar{X})\right\|_F^2 = (\mu e e^T - \bar{X})_+ = (\bar{X} - \mu e e^T)_-,
\tag{15}
$$

since the (Euclidean) projection of a symmetric matrix $M$ to the positive semidefinite cone is $M_+$ [3, 22]. Thus, to solve the dual, it suffices to minimize the univariate convex function

$$
g(\mu) = \mu + \frac{1}{2}\left\|\Lambda(\mu) - (\mu e e^T - \bar{X})\right\|_F^2 = \mu + \frac{1}{2}\left\|(\bar{X} - \mu e e^T)_+\right\|_F^2 = \mu + \frac{1}{2}\sum_{i=1}^n [\lambda_i(\bar{X} - \mu e e^T)]_+^2.
\tag{16}
$$

This, in turn, reduces to finding a root of the derivative of $g$, since the second term is continuously differentiable [23] and $\mu$ is unconstrained. The derivative, denoted by $f$ hereafter, has a closed form:

$$
f(\mu) = 1 - e^T(\bar{X} - \mu e e^T)_+ e,
\tag{17}
$$

which is monotone nondecreasing since $g$ is convex. From $\bar{X}_+ = \bar{X} + \bar{X}_-$ and $e^T \bar{X} e = 1$, it follows that $e^T \bar{X}_+ e = 1 + e^T \bar{X}_- e \geq 1$ hence $f(0) \leq 0$. Since $g(\mu) \geq \mu$, we see $f(\mu) > 0$ for sufficiently large $\mu$ and a root $\mu^\star$ of $f$ exists. Further, as $\mu^\star$ minimizes $g$, we have $\frac{1}{2}\|\bar{X}_+\|_F^2 = g(0) \geq g(\mu^\star) \geq \mu^\star$ and $\mu^\star \in [0, \|\bar{X}\|_F^2/2]$.

The remaining dual solution is $\Lambda^\star = \Lambda(\mu^\star) = (\bar{X} - \mu^\star e e^T)_-$. From this, relation (14), and construction (11), the sought projection $P_C(X, y) = (V^\star, w^\star)$ is evaluated. From the Moreau decomposition it is clear that $\mathrm{prox}_\phi(X, y) = (X - V^\star, y - w^\star)$. In fact,

$$
\Lambda^\star = U^\star - (\bar{X} - \mu^\star e e^T) = \begin{bmatrix} X - V^\star & -\frac{1}{\sqrt{2}}(y - w^\star) \\ \frac{1}{\sqrt{2}}(y - w^\star)^T & \mu^\star \end{bmatrix}.
$$

Thus $\mathrm{prox}_\phi(X, y)$ can be directly obtained from $\Lambda^\star$. Furthermore, $\mu^\star$ is related with $\Lambda^\star$ by

$$
\mu^\star = e^T \Lambda^\star e.
$$

The findings so far are summarized as the following theorem.

**Theorem 1.** *Suppose $(\Omega^\star, \eta^\star) = \mathrm{prox}_\phi(X, y)$. Construct a block matrix $\bar{X} \in \mathbb{S}^n$ as in (11). If $\mu^\star$ is a nonnegative root, lying in $[0, \|\bar{X}\|_F^2/2]$, of the univariate, monotone nondecreasing function $f(\mu)$ in (17), then for the positive semidefinite matrix*

$$
\Lambda^\star = (\bar{X} - \mu^\star e e^T)_- = \begin{bmatrix} \Lambda_{11}^\star & \lambda_{12}^\star \\ \lambda_{12}^{\star T} & \lambda_{22}^\star \end{bmatrix}, \quad \Lambda_{11}^\star \in \mathbb{S}^p,
$$

*we have $\Omega^\star = \Lambda_{11}^\star$ and $\eta^\star = -\sqrt{2}\lambda_{12}^\star$. Furthermore, there holds $\mu^\star = \lambda_{22}^\star$.*

**Remark 1.** *In fact the root $\mu^\star$ of $f$ is unique. This is proved in Theorem 2 in the next section, since showing the uniqueness requires further analysis of the function $f$.*

## 3 Quadratically convergent Newton algorithm

Utilizing the connection to the matrix nearness problem established in the previous section, in this section we develop a Newton algorithm for finding the unique root of the function $f$ in (17) and show that it converges quadratically. While bisection will converge linearly, since the proximity operator, found by a closed form calculation from the root, is evaluated iteratively in proximal algorithms such as PDHG (6), a faster and more accurate root-finding method is desirable. Unfortunately the function $f$ is not differentiable everywhere [4]. In most situations, Newton's algorithm would not be applicable. Nevertheless, it can be shown that the function $f$ is *strongly semismooth*, from which a quadratically convergent Newton algorithm can be devised. A similar approach can be found for the nearest correlation matrix problem, another instance of the matrix nearness problem [2, 28].

We begin with relevant definitions.

**Definition 1** (Clarke's generalized Jacobian [8]). *For a function $F : \mathbb{R}^m \to \mathbb{R}^l$ that is locally Lipschitz around $\boldsymbol{x} \in \mathbb{R}^m$, Clarke's generalized Jacobian is*

$$\partial F(\boldsymbol{x}) = \mathbf{conv}\{\lim_k \nabla F(\boldsymbol{x}^k) : \boldsymbol{x}^k \to \boldsymbol{x}, \ \boldsymbol{x}^k \in D_F(\boldsymbol{x})\}, \ D_F(\boldsymbol{x}) = \{\boldsymbol{y} : F \text{ is differentiable at } \boldsymbol{y}\},$$

*where $\mathbf{conv}$ denotes the convex hull operation and $\nabla F(\boldsymbol{y})$ denotes the Jacobian of $F$ at $\boldsymbol{y}$.*

If $F$ is real-valued and convex, then the Clarke generalized Jacobian reduces to the usual convex subdifferential. The set $\partial F(\boldsymbol{x})$ is compact and the set-valued map $\partial F$ is upper semicontinuous: if $\boldsymbol{x}_k \to \boldsymbol{x}$ and $\boldsymbol{y}_k \to \boldsymbol{y}$ for $\boldsymbol{y}_k \in \partial F(\boldsymbol{x}_k)$, then $\boldsymbol{y} \in \partial F(\boldsymbol{x})$.

**Definition 2** (semismoothness [28, 29]). *Function $F : \mathbb{R}^m \to \mathbb{R}^l$ is semismooth at $\boldsymbol{x} \in \mathbb{R}^m$ if it is locally Lipschitz, directionally differentiable at $\boldsymbol{x}$, and for any $\boldsymbol{V} \in \partial F(\boldsymbol{x} + \boldsymbol{h})$, we have $F(\boldsymbol{x} + \boldsymbol{h}) - F(\boldsymbol{x}) - \boldsymbol{V}\boldsymbol{h} = o(\|\boldsymbol{h}\|)$. A semismooth function $F$ is strongly semismooth at $\boldsymbol{x}$ if it is semismooth at $\boldsymbol{x}$ and for any $\boldsymbol{V} \in \partial F(\boldsymbol{x} + \boldsymbol{h})$, we have $F(\boldsymbol{x} + \boldsymbol{h}) - F(\boldsymbol{x}) - \boldsymbol{V}\boldsymbol{h} = O(\|\boldsymbol{h}\|^2)$.*

If we let $\phi(x) = x_+$ for $x \in \mathbb{R}$ and $\phi^\square(\boldsymbol{X}) = \boldsymbol{P} \operatorname{diag}(\phi(\lambda_1), \dots, \phi(\lambda_n))\boldsymbol{P}^T = \boldsymbol{X}_+$ for $\boldsymbol{X} = \boldsymbol{P} \operatorname{diag}(\lambda_1, \dots, \lambda_n)\boldsymbol{P}^T \in \mathbb{S}^n$ where $\boldsymbol{P}$ satisfies $\boldsymbol{P}^T\boldsymbol{P} = \boldsymbol{P}\boldsymbol{P}^T = \boldsymbol{I}$, then it is clear from equation (17) that $f(\mu) = 1 - \boldsymbol{e}^T \phi^\square(\boldsymbol{C}(\mu))\boldsymbol{e}$, for $\boldsymbol{C}(\mu) = \bar{\boldsymbol{X}} - \mu \boldsymbol{e}\boldsymbol{e}^T$.

Function $\phi^\square$ is 1-Lipschitz and strongly semismooth everywhere on $\mathbb{S}^n$ [7, 31], from which it follows that $f$ is also 1-Lipschitz and strongly semismooth everywhere on $\mathbb{R}$. A general result on the Newton methods for semismooth functions ensures that the Newton update $\mu_{k+1} = \mu_k - f(\mu_k)/v_k$ with $v_k \in \partial f(\mu_k) \subset \mathbb{R}$ converges *quadratically* to a root $\mu^\star$ of $f$, provided that $v \neq 0$ for all $v \in \partial f(\mu^\star)$ and the starting point $\mu_0$ is sufficiently close to $\mu^\star$ [28, Thm. 2.1]. Thus to establish locally quadratic convergence, it suffices to show that any $v \in \partial f(\mu^\star)$ is nonzero. In fact we can say more, including the uniqueness of the root:

**Theorem 2.** *Function $f(\mu)$ of Theorem 1 has a unique root $\mu^\star$. Each element $v \in \partial f(\mu^\star)$ is positive.*

The proof of Theorem 2 is technical and lengthy, and is deferred to the Supplement.

In order to ensure global convergence, we consider Algorithm 1, which is similar in spirit to the guarded Newton method considered by Boyd and Xiao [4, §3.4] for a *smooth* function.

---

**Algorithm 1** Guarded Newton

---

  *Input:* Starting value $\mu_0 \in [0, \|\bar{\boldsymbol{X}}\|_F^2/2]$
  Initial interval: $(l, u) \leftarrow (0, \|\bar{\boldsymbol{X}}\|_F^2/2)$; index $k \leftarrow 0$
  **repeat**
     Select $v_k \in \partial f(\mu_k)$
     **if** $v_k > 0$ **then**
        Pure Newton step: $\Delta \mu_k \leftarrow -f(\mu_k)/v_k$
     **else**
        Gradient step: $\Delta \mu_k \leftarrow -f(\mu_k)/(v_k + |f(\mu_k)|)$
     **end if**
     Project onto guard interval: $\mu_{k+1} \leftarrow P_{[l,u]}(\mu + \Delta\mu_k)$
     Update guard interval: $u \leftarrow \mu_{k+1}$ if $f(\mu_{k+1}) > 0$; otherwise $l \leftarrow \mu_{k+1}$
     $k \leftarrow k + 1$
  **until** convergence
  **return** $\mu_{k+1}$

---

Note, if $\Delta\mu_k$ is replaced by $(u+l)/2 - \mu_k$, then Algorithm 1 reduces to bisection. Global convergence of the Newton algorithm is established as follows.

**Theorem 3.** *The sequence $\{\mu_k\}$ generated by Algorithm 1 converges to the unique root $\mu^\star$ of the function $f$ of Theorem 1. Convergence of $\{\mu_k\}$ is asymptotically quadratic.*

*Proof.* For each $k$, $s_k = \mu_{k+1} - \mu_k$ is a descent direction of the objective function $g$. Since $g$ is bounded below and $\mu_k$ is bounded within $[0, \|\bar{\boldsymbol{X}}\|_F^2/2]$, a standard result on the convergence of algorithms involving descent steps and Lipschitzian gradients [35, 36] asserts that $\lim_k f(\mu_k) = 0 = f(\mu^\star)$. (Recall $f$ is the derivative of $g$.) Let $z_k = f(\mu_k)$. Clearly $\lim_k z_k = 0$. From Theorem 2, for any $v \in \partial f(\mu^\star)$ we have $v > 0$. Then Clarke's inverse function theorem [8, Thm. 7.1.1] entails that there is a Lipschitzian inverse function $f^{-1}$ on some neighborhood of $\mu^\star$. Thus for sufficiently large $k$, we have $\mu_k = f^{-1}(z_k) \to f^{-1}(0) = \mu^\star$.

Combining the global 1-Lipschitzness and monotonicity of $f$, and Definition 1, we see $0 \le v_k \le 1$ for all $k$. Thus by the Bolzano-Weirstrauss Theorem, $\{v_k\}$ has a convergent subsequence $\{v_{k_l}\}$, whose limit is a cluster point of $\{v_k\}$. Conversely, for any cluster point $v^*$ of $\{v_k\}$, there is a subsequence $\{v_{k_l}\}$ converging to $v^\star$. Then, since $\mu_{k_l} \to \mu^\star$, by the upper semicontinuity of the map $\partial f$, we have $v_{k_l} \to v^* \in \partial f(\mu^\star)$. From Theorem 2, $v^* > 0$. In particular, $0 < \liminf_k v_k \in \partial f(\mu^\star)$. Therefore, for sufficiently large $k$, there exists $\gamma > 0$ such that $v_k \ge \gamma$. For such $k$, $\Delta\mu_k = -f(\mu_k)/v_k$ and

$$\begin{aligned}
|\mu_k + \Delta\mu_k - \mu^\star| &= |\mu_k + [(f(\mu_k) + v_k\Delta\mu_k) - f(\mu_k)]/v_k - \mu^\star| \\
&\le |\mu_k - \mu^\star - f(\mu_k)/v_k| + |(f(\mu_k) + v_k\Delta\mu_k)/v_k| \\
&\le \frac{1}{\gamma}|(f(\mu_k) - f(\mu^\star)) - v_k(\mu_k - \mu^\star)| + 0 = O(|\mu_k - \mu^\star|^2).
\end{aligned}$$

The second inequality uses $f(\mu^\star) = 0$ and the final equality is from the strong semismoothness of $f$.

Let $\tilde{\mu}_{k+1} = \mu_k + \Delta\mu_k$. For each $k$, we have $l \le \mu^\star \le u$. For sufficiently large $k$, either $u = \mu_k$ or $l = \mu_k$. If $u = \mu_k$, then $f(\mu_k) > 0$ and $\tilde{\mu}_{k+1} = \mu_k - f(\mu_k)/v_k < \mu_k$. There are three possible orderings of $\tilde{\mu}_{k+1}$ with respect to $l$, $u$, and $\mu^\star$. If $l \le \mu^\star \le \tilde{\mu}_{k+1} \le \mu_k = u$ or $l \le \tilde{\mu}_{k+1} \le \mu^\star \le \mu_k = u$, then $\mu_{k+1} = \tilde{\mu}_{k+1}$. Otherwise $\tilde{\mu}_{k+1} \le l \le \mu^\star \le \mu_k = u$, yielding $\mu_{k+1} = l$. In all cases, we obtain

$$|\mu_{k+1} - \mu^\star| \le |\tilde{\mu}_{k+1} - \mu^\star| = |\mu_k + \Delta\mu_k - \mu^\star| \le O(|\mu_k - \mu^\star|^2).$$

A parallel argument for the case $l = \mu_k$ results in the same conclusion. □

Algorithm 1 needs a $v_k \in \partial f(\mu_k)$. The following theorem, proved in the Supplement, presents a closed form. For any $\boldsymbol{\lambda} = (\lambda_1, \ldots, \lambda_n)$, denote by $\phi^{[1]}(\boldsymbol{\lambda})$ the $n \times n$ symmetric matrix with entries

$$\phi_{ij}^{[1]}(\boldsymbol{\lambda}) = \begin{cases} \frac{\phi(\lambda_i) + \phi(\lambda_j)}{|\lambda_i| + |\lambda_j|}, & \lambda_i \ne 0 \text{ or } \lambda_j \ne 0, \\ 0, & \lambda_i = \lambda_j = 0. \end{cases} \tag{18}$$

**Theorem 4.** *For a spectral decomposition of $\boldsymbol{C}(\mu) = \bar{\boldsymbol{X}} - \mu\boldsymbol{e}\boldsymbol{e}^T$, i.e., $\boldsymbol{C}(\mu) = \boldsymbol{P}\operatorname{diag}(\lambda_1, \ldots, \lambda_n)\boldsymbol{P}^T$ with $\boldsymbol{P}^T\boldsymbol{P} = \boldsymbol{P}\boldsymbol{P}^T = \boldsymbol{I}$, set $\boldsymbol{\lambda} = (\lambda_1, \ldots, \lambda_n)$. Denote by $\circ$ element-wise matrix multiplication. Then,*

$$v = \boldsymbol{e}^T\boldsymbol{P}(\phi^{[1]}(\boldsymbol{\lambda}) \circ (\boldsymbol{P}^T\boldsymbol{e}\boldsymbol{e}^T\boldsymbol{P}))\boldsymbol{P}^T\boldsymbol{e} \in \partial f(\mu).$$

**Remark 2.** *In [2, 28], the constraint is that all diagonal entries of $\boldsymbol{U}$ in SDP (12) are 1. Rather surprisingly, our simpler constraint makes the perturbation analysis of the spectral decomposition of $\boldsymbol{C}(\mu)$ more difficult (see Lemma A.2 and Sect. A.3 of the Supplement) than [28, Lemma 3.4].*

**Computational concerns** Algorithm 1 requires a full spectral decomposition of $\boldsymbol{C}(\mu)$, which costs around $10n^3$, for each iteration. Since $\boldsymbol{C}(\mu)$ is a symmetric rank-1 perturbation of $\bar{\boldsymbol{X}}$, precomputed spectral decomposition of $\bar{\boldsymbol{X}}$ can be efficiently updated using the deflation technique [5, 13].

## 4 Empirical results

### 4.1 Performance of the Newton method

We begin this section with assessing the performance of the proposed Newton method (Algorithm 1) for the proximity operator $\mathbf{prox}_\phi$. Since the problem of computing this operator is SDP with dual

(12), we compared Algorithm 1 with a commercial SDP solver MOSEK [26] as well as bisection. The algorithm was implemented in the Julia programming language on a standard laptop (Macbook Pro 2019, i5@2.4GHz, 16GB RAM), and MOSEK was invoked via its Julia interface Convex.jl [33]. Results under several performance measures are reported in Table 1, averaged over 100 randomly sampled Gaussian input points external to the cone (8). (Within the parentheses are standard deviations.) Runtime is assessed via number of iterations ("Iters") as well as elapsed time in seconds ("Secs") until convergence. The "Obj" and "KKT" measures respectively refer to the value of the objective function (16) and the absolute value of its derivative (17) at convergence; convergence was declared when the KKT measure was $< 10^{-8}$.

Our results clearly reveal the quadratic convergence behavior of Algorithm 1: it terminated within 8 iterations. Until the point that MOSEK failed to scale ($p < 500$), our Newton method was orders of magnitude faster and more accurate (in terms of KKT) than the commercial solver. Although bisection was also faster than MOSEK, it was slower and in general orders of magnitude less accurate than Newton. A typical convergence plot is shown in Fig. 1. The speed persisted for larger $p$s: it took less than 5 seconds to solve a problem of size $2000 \times 2000$.

Table 1: Average performance of the Newton method

| $p$ | Method | Iters | Secs | KKT | Obj |
|---|---|---|---|---|---|
| 10 | MOSEK | – | 0.007020 (0.0009176) | 8.599e-6 (8.273e-6) | 3.9326 (1.659) |
| | Bisection | 27.30 (1.059) | 0.0002300 (2.627e-5) | 5.086e-9 (3.172e-9) | 3.9326 (1.659) |
| | Newton | 4.900 (0.5676) | 0.0001568 (4.514e-5) | 9.719e-10 (2.018e-9) | 3.9326 (1.659) |
| 30 | MOSEK | – | 0.1285 (0.08261) | 8.512e-6 (9.167e-6) | 16.262 (3.781) |
| | Bisection | 28.40 (0.6992) | 0.001044 (4.624e-5) | 4.015e-9 (3.040e-9) | 16.262 (3.781) |
| | Newton | 5.900 (0.3162) | 0.0005461 (3.596e-5) | 1.884e-10 (5.957e-10) | 16.262 (3.781) |
| 50 | MOSEK | – | 0.5566 (0.07094) | 2.114e-6 (3.989e-5) | 26.762 (5.537) |
| | Bisection | 28.70 (0.6749) | 0.002824 (0.0003610) | 5.678e-9 (2.732e-9) | 26.762 (5.537) |
| | Newton | 6.000 (0.0000) | 0.001192 (5.717e-5) | 1.725-11 (2.919e-11) | 26.762 (5.537) |
| 100 | MOSEK | – | 13.60 (3.9351) | 3.071e-6 (2.955e-6) | 60.299 (9.586) |
| | Bisection | 29.00 (1.563) | 0.009690 (0.001674) | 2.793e-9 (1.695e-9) | 60.299 (9.586) |
| | Newton | 6.000 (0.0000) | 0.006363 (0.006630) | 2.574-9 (1.980e-9) | 60.299 (9.586) |
| 500 | MOSEK | – | – | – | – |
| | Bisection | 29.10 (2.0790) | 0.3001 (0.01540) | 4.590-9 (3.138e-9) | 319.86 (19.80) |
| | Newton | 7.000 (0.0000) | 0.1166 (0.003669) | 8.299-10 (3.912e-10) | 319.86 (19.80) |
| 1000 | MOSEK | – | – | – | – |
| | Bisection | 30.20 (1.3166) | 1.873 (0.09942) | 4.240e-9 (2.810e-9) | 661.19 (26.94) |
| | Newton | 8.000 (0.0000) | 0.8073 (0.03513) | 1.417-14 (5.679e-15) | 661.19 (26.94) |
| 2000 | MOSEK | – | – | – | – |
| | Bisection | 29.50 (3.1002) | 11.60 (1.048) | 3.577e-9 (2.634e-9) | 1356.36 (47.93) |
| | Newton | 8.000 (0.0000) | 4.763 (0.03273) | 3.621-11 (1.961e-11) | 1356.36 (47.93) |

## 4.2 Applications to proximal algorithms

We then applied operator $\mathbf{prox}_\phi$ to the PDHG algorithm (6) for solving the three learning problems introduced in Section 1. The results are summarized in Table 2. Detailed derivation of the PDHG iteration, setup, and convergence criteria for each problem appear in the Supplement.

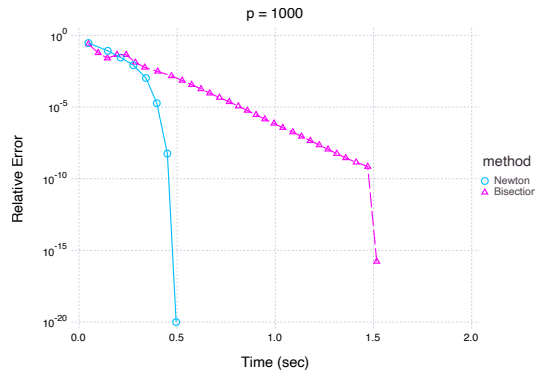

Figure 1: Convergence of semismooth Newton and bisection methods.

**Heteroskedastic scaled lasso.** Problem (4) can be reformulated as a SDP. Hence MOSEK was used as a benchmark. An $N \times p$ data matrix $\boldsymbol{X}$ was sampled from independent Gaussian. Response $\boldsymbol{y}$ was corrupted by correlated noise with compound symmetry. The "Numvars" measure indicates the number of scalar variables fed to $\mathbf{prox}_\phi$ (note $\boldsymbol{\Omega} \in \mathbb{S}^N$). MOSEK failed to scale for $N > 200$. For small $n$, PDHG was as accurate as MOSEK ("Relerr" measures the relative error of the PHDG solution to the MOSEK solution in the Frobenius norm.) The five leading eigenvalues of the computed solution $\boldsymbol{\Omega}$ is given. As the sample size $N$ grows the low-rank structure of the error covariance matrix appears to be recovered. Study of statistical properties of model (4) is not the scope of this paper.

**Gaussian joint likelihood estimation** To the objective (2), unit variance constraints were imposed on the first five diagonal components of $\boldsymbol{\Sigma}$. This problem could not be solved with MOSEK. An $N \times p$ data matrix $\boldsymbol{X}$ was sampled from a zero-mean multivariate Gaussian with a compound symmetric covariance matrix, from which sufficient statistics $\boldsymbol{S}$ and $\bar{\boldsymbol{\mu}}$ were fed to PDHG. Constraint violation was measured by the excess from 1 in the first five diagonal entries of estimated covariance matrix. PDHG iterates did not converge after 50000 iterations for $p \geq 500$ (objective value converged, though), while constraint violations are relatively small given the difficulty of the problem due to its size. For comparison, the largest of the first five diagonal entries of the sample covariance matrix $\boldsymbol{S} - \bar{\boldsymbol{\mu}}\bar{\boldsymbol{\mu}}^T$ is also provided.

**Graphical model.** The PDHG iteration for problem (3) (with an $\ell_1$ penalty) entails a dual variable of size $Np \times Np$ fed to $\mathbf{prox}_\phi$ (see Supplement). A small dimension $p = 50$ of precision matrix $\boldsymbol{\Omega}$ readily yields a $1500 \times 1500$ dimensional matrix variable (with $N = 30$). Despite this drawback, PDHG is a rare method that minimizes the $\ell_1$-penalized pseudolikelihood (3) *globally* with a convergence guarantee. While there are many pseudolikelihood-based graphical model selection methods [15, 19, 25, 27, 30], they either alter the objective or reparameterize it into a nonconvex problem [19]. Among these, the symmetric lasso [15] employs the unaltered objective, hence was compared. Clearly the symmetric lasso results in a suboptimal solution with larger objective values ("Obj-sym") and 7–8% of relative errors; "'NZ" refers to the fraction of nonzero components in the estimated $\boldsymbol{\Omega}$.

Table 2: Applications to proximal algorithms

|  | $N$ | $p$ | Numvars | Iters | Obj-PDHG | Obj-Mosek | Relerr | Leading eigenvalues |
|---|---|---|---|---|---|---|---|---|
| Scaled lasso | 50 | 20 | 1245 | 8001 | 3.41240 | 3.41240 | 0.0009726 | (21.72, 6.299e-7, 3.335e-9, 7.991e-10) |
|  | 100 | 20 | 4970 | 7513 | 2.65602 | 2.65602 | 0.001270 | (23.48, 1.404e-5, 8.95e-7, 1.913e-7) |
|  | 200 | 20 | 19920 | 11800 | 3.20913 | 3.20913 | 0.001666 | (41.02, 4.015e-4, 5.093e-5, 4.530e-5) |
|  | 300 | 20 | 44870 | 9188 | 3.61066 | – | – | (59.46, 0.03216, 0.01126, 0.01021) |
|  | 400 | 20 | 79820 | 15400 | 6.33631 | – | – | (123.4, 0.05013, 0.04788, 0.03578) |
|  | 500 | 20 | 124800 | 13270 | 5.12763 | – | – | (112.8, 0.09574, 0.06423, 0.05875) |

|  | $N$ | $p$ | Numvars | Iters | Obj-PDHG | Constraint violation | Largestdiag |
|---|---|---|---|---|---|---|---|
| Gaussian Joint MLE | 30 | 50 | 1275 | 4378 | -55.25 | (9.389e-6, 5.236e-5, 0, 0, 0) | 1.217 |
|  | 60 | 100 | 5050 | 14510 | -286.55 | (1.583e-5, 0, 0, 0, 6.174e-6) | 1.252 |
|  | 100 | 200 | 20100 | 42470 | -3351.04 | (4.075e-5, 0, 0, 0, 3.238e-5) | 1.261 |
|  | 300 | 500 | 125200 | 50000 | -7279.68 | (0, 0, 0, 0.0002229, 0) | 1.093 |
|  | 500 | 1000 | 500500 | 50000 | -12671.03 | (0.02444, 0, 0.002228, 0.003762, 0.007134) | 1.120 |

|  | $N$ | $p$ | Numvars | Iters | Obj-PDHG | Obj-sym | NZ-PDHG | NZ-sym | Relerr |
|---|---|---|---|---|---|---|---|---|---|
| Graphical model selection | 10 | 10 | 4950 | 1255 | -6.2803 | -6.2510 | 0.2600 | 0.2600 | 0.0777 |
|  | 20 | 30 | 179700 | 1240 | -18.8627 | -18.7895 | 0.0600 | 0.0600 | 0.0868 |
|  | 30 | 50 | 1124250 | 1069 | -33.9688 | -33.8675 | 0.0256 | 0.0264 | 0.0793 |

## 5   Discussion

Given the significance of the multivariate Gaussian in machine learning and statistics, enlarging the class of tractable estimation problems is important and useful for both communities, let alone other problems discussed in this paper. Joint estimation of Gaussian natural parameters under constraints has not received much attention, and it appears that a part of the reason is the lack of practical optimization algorithms. Our contributions on the matrix perspective function enable proximal methods to embrace previously intractable optimization problems arising from important learning tasks. Further developments, e.g, acceleration and scale up of PDHG, are natural next steps.

## Broader Impact

Not applicable.

## Acknowledgments and Disclosure of Funding

This work was supported by the National Research Foundation of Korea (NRF) grant funded by the Korea government (MSIT) (No. 2019R1A2C1007126). There are no competing interests.

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
