[Supplementary Material]

# Supplement: Proximity Operator of the Matrix Perspective Function and its Applications

**Joong-Ho Won**
Department of Statistics
Seoul National University
wonj@stats.snu.ac.kr

## A  Proofs

### A.1  A key lemma

Proofs of both Theorems 2 and 4 are based on the following key lemma, Lemma A.1. Recall that $\phi(x) = x_+$ for $x \in \mathbb{R}$ and $\phi^\square(\boldsymbol{X}) = \boldsymbol{P}\operatorname{diag}(\phi(\lambda_1),\ldots,\phi(\lambda_n))\boldsymbol{P}^T = \boldsymbol{X}_+$ for $\boldsymbol{X} = \boldsymbol{P}\operatorname{diag}(\lambda_1,\ldots,\lambda_n)\boldsymbol{P}^T \in \mathbb{S}^n$ where $\boldsymbol{P}$ satisfies $\boldsymbol{P}^T\boldsymbol{P} = \boldsymbol{P}\boldsymbol{P}^T = \boldsymbol{I}$. For any $\boldsymbol{\lambda} = (\lambda_1,\ldots,\lambda_n)$, $\phi^{[1]}(\boldsymbol{\lambda})$ is the $n \times n$ symmetric matrix with $(i,j)$ entry

$$\phi_{ij}^{[1]}(\boldsymbol{\lambda}) = \begin{cases} \frac{\phi(\lambda_i)+\phi(\lambda_j)}{|\lambda_i|+|\lambda_j|}, & \lambda_i \neq 0 \text{ or } \lambda_j \neq 0, \\ 0, & \lambda_i = \lambda_j = 0. \end{cases} \tag{18}$$

Also recall that $\boldsymbol{C}(\mu) = \bar{\boldsymbol{X}} - \mu \boldsymbol{e}\boldsymbol{e}^T$ so that $f(\mu) = g'(\mu) = 1 - \boldsymbol{e}^T\phi^\square(\boldsymbol{C}(\mu))\boldsymbol{e}$. Lemma A.1 provides a closed-form expression of the derivative of $f(\mu)$ when it exists, in terms of the matrix function (18).

**Lemma A.1.** *Function $f$ is differentiable at $\mu$ if and only if $\boldsymbol{e} \in \mathcal{N}(\boldsymbol{C}(\mu))^\perp$. In this case, the derivative is*

$$f'(\mu) = \boldsymbol{e}^T\boldsymbol{P}(\phi^{[1]}(\boldsymbol{\lambda}) \circ (\boldsymbol{P}^T\boldsymbol{e}\boldsymbol{e}^T\boldsymbol{P}))\boldsymbol{P}^T\boldsymbol{e}, \tag{A.1}$$

*for any $\boldsymbol{\lambda} = (\lambda_1,\ldots,\lambda_n)^T$ and $\boldsymbol{P}$ satisfying $\boldsymbol{C}(\mu) = \boldsymbol{P}\operatorname{diag}(\lambda_1,\ldots,\lambda_n)\boldsymbol{P}^T$, $\boldsymbol{P}^T\boldsymbol{P} = \boldsymbol{P}\boldsymbol{P}^T = \boldsymbol{I}$.*

To prove this lemma, we begin by recalling the definition of directional derivatives.

**Definition A.1** (Directional derivative). *For a function $F : \mathbb{R}^m \to \mathbb{R}^l$ and $\boldsymbol{x}, \boldsymbol{h} \in \mathbb{R}^m$, the directional derivative of $F$ at $\boldsymbol{X}$ along $\boldsymbol{h}$ is defined and denoted by*

$$F'(\boldsymbol{x}; \boldsymbol{h}) = \lim_{t\downarrow 0} \frac{F(\boldsymbol{x}+t\boldsymbol{h}) - F(\boldsymbol{x})}{t}$$

*if the limit exists. The $F$ is called directionally differentiable at $\boldsymbol{x}$ if $F'(\boldsymbol{x}; \boldsymbol{h})$ exists for all $\boldsymbol{h} \in \mathbb{R}^m$.*

If $F$ is differentiable at $\boldsymbol{x}$ with Jacobian $\nabla F(\boldsymbol{x}) \in \mathbb{R}^{l \times m}$, then $F'(\boldsymbol{x}; \boldsymbol{h}) = \nabla F(\boldsymbol{x})\boldsymbol{h}$.

For index sets $J, K \subset \{1,\ldots,n\}$ and matrix $\boldsymbol{M} \in \mathbb{S}^n$, let $\boldsymbol{M}_{JK}$ be the submatrix of $\boldsymbol{M}$ constructed from the rows in $J$ and the columns in $K$. The following lemma can be deduced from Sun and Sun [2002, Theorem 4.7]:

**Lemma A.2.** *Function $\phi^\square$ is directionally differentiable at any $\boldsymbol{X} \in \mathbb{S}^n$. Its directional derivative along $\boldsymbol{H} \in \mathbb{S}^n$ is*

$$(\phi^\square)'(\boldsymbol{X}; \boldsymbol{H}) = \boldsymbol{P} \begin{bmatrix} \phi_{KK}^{[1]}(\boldsymbol{\lambda}) \circ \tilde{\boldsymbol{H}}_{KK} & \phi_{KJ}^{[1]}(\boldsymbol{\lambda}) \circ \tilde{\boldsymbol{H}}_{KJ} \\ \phi_{JK}^{[1]}(\boldsymbol{\lambda}) \circ \tilde{\boldsymbol{H}}_{JK} & [\tilde{\boldsymbol{H}}_{JJ}]_+ \end{bmatrix} \boldsymbol{P}^T$$

*where $\boldsymbol{X} = \boldsymbol{P}\operatorname{diag}(\lambda_1,\ldots,\lambda_n)\boldsymbol{P}^T \in \mathbb{S}^n$ with $\boldsymbol{P}$ satisfying $\boldsymbol{P}^T\boldsymbol{P} = \boldsymbol{P}\boldsymbol{P}^T = \boldsymbol{I}$, $K = \{i \in \{1,\ldots,n\} : \lambda_i \neq 0\}$, $J = \{i \in \{1,\ldots,n\} : \lambda_i = 0\}$, and $\tilde{\boldsymbol{H}} = \boldsymbol{P}^T\boldsymbol{H}\boldsymbol{P}$. Furthermore, $\phi^\square$ is differentiable at $\boldsymbol{X}$ if and only if $\boldsymbol{X}$ is nonsingular, i.e., $J = \emptyset$.*

Now we can prove the lemma:

*Proof of Lemma A.1.* Suppose $f$ is differentiable at $\mu$. Then if $\boldsymbol{C}(\mu)$ is nonsingular, $\mathcal{N}(\boldsymbol{C}(\mu)) = \{\boldsymbol{0}\}$ and $\boldsymbol{e} \in \mathcal{N}(\boldsymbol{C}(\mu))^\perp$. If $\boldsymbol{C}(\mu)$ is singular, then the two one-sided limits

$$\lim_{t \downarrow 0} \frac{f(\mu + t) - f(\mu)}{t} \quad \text{and} \quad \lim_{t \downarrow 0} \frac{f(\mu - t) - f(\mu)}{-t}$$

must coincide. The first limit is equal to

$$-\boldsymbol{e}^T \left( \lim_{t \downarrow 0} \frac{\phi^\square(\boldsymbol{C}(\mu + t)) - \phi^\square(\boldsymbol{C}(\mu))}{t} \right) \boldsymbol{e} = -\boldsymbol{e}^T \left( \lim_{t \downarrow 0} \frac{\phi^\square(\boldsymbol{C}(\mu) - t\boldsymbol{e}\boldsymbol{e}^T) - \phi^\square(\boldsymbol{C}(\mu))}{t} \right) \boldsymbol{e}$$

$$= -\boldsymbol{e}^T (\phi^\square)'(\boldsymbol{C}(\mu); -\boldsymbol{e}\boldsymbol{e}^T)\boldsymbol{e}$$

$$= \boldsymbol{e}^T \boldsymbol{P} \begin{bmatrix} \phi_{KK}^{[1]}(\boldsymbol{\lambda}) \circ (\boldsymbol{P}^T \boldsymbol{e}\boldsymbol{e}^T \boldsymbol{P})_{KK} & \phi_{KJ}^{[1]}(\boldsymbol{\lambda}) \circ (\boldsymbol{P}^T \boldsymbol{e}\boldsymbol{e}^T \boldsymbol{P})_{KJ} \\ \phi_{JK}^{[1]}(\boldsymbol{\lambda}) \circ (\boldsymbol{P}^T \boldsymbol{e}\boldsymbol{e}^T \boldsymbol{P})_{JK} & -[(\boldsymbol{P}^T \boldsymbol{e}\boldsymbol{e}^T \boldsymbol{P})_{JJ}]_+ \end{bmatrix} \boldsymbol{P}^T \boldsymbol{e}$$

by Lemma A.2, for a spectral decomposition of $\boldsymbol{C}(\mu) = \boldsymbol{P} \operatorname{diag}(\lambda_1, \ldots, \lambda_n)\boldsymbol{P}^T$ satisfying the conditions of the lemma. Likewise, the second limit equals

$$\boldsymbol{e}^T (\phi^\square)'(\boldsymbol{C}(\mu); \boldsymbol{e}\boldsymbol{e}^T)\boldsymbol{e} = \boldsymbol{e}^T \boldsymbol{P} \begin{bmatrix} \phi_{KK}^{[1]}(\boldsymbol{\lambda}) \circ (\boldsymbol{P}^T \boldsymbol{e}\boldsymbol{e}^T \boldsymbol{P})_{KK} & \phi_{KJ}^{[1]}(\boldsymbol{\lambda}) \circ (\boldsymbol{P}^T \boldsymbol{e}\boldsymbol{e}^T \boldsymbol{P})_{KJ} \\ \phi_{JK}^{[1]}(\boldsymbol{\lambda}) \circ (\boldsymbol{P}^T \boldsymbol{e}\boldsymbol{e}^T \boldsymbol{P})_{JK} & [(\boldsymbol{P}^T \boldsymbol{e}\boldsymbol{e}^T \boldsymbol{P})_{JJ}]_+ \end{bmatrix} \boldsymbol{P}^T \boldsymbol{e}.$$

Let $\boldsymbol{P}^T \boldsymbol{e} = [\boldsymbol{q}_K, \boldsymbol{q}_J]^T = \boldsymbol{q}$ where $\boldsymbol{q}_K \in \mathbb{R}^{|K|}$ and $\boldsymbol{q}_J \in \mathbb{R}^{|J|}$. Note $J \neq \emptyset$ since $\boldsymbol{C}(\mu)$ is singular. Then the two limits are equal if and only if $\boldsymbol{q}_J^T [(\boldsymbol{q}\boldsymbol{q}^T)_{JJ}]_+ \boldsymbol{q}_J = 0$. It is immediate to see that $(\boldsymbol{q}\boldsymbol{q}^T)_{JJ} = \boldsymbol{q}_J \boldsymbol{q}_J^T \succeq \boldsymbol{0}$, hence $\boldsymbol{q}_J^T [(\boldsymbol{q}\boldsymbol{q}^T)_{JJ}]_+ \boldsymbol{q}_J = \|\boldsymbol{q}_J\|^4$. This implies $\boldsymbol{q}_J = \boldsymbol{0}$. Finally, observe that $\boldsymbol{q}_J = \boldsymbol{P}_J^T \boldsymbol{e}$ where the columns of $\boldsymbol{P}_J$ span $\mathcal{N}(\boldsymbol{C}(\mu))$. Thus the condition $\boldsymbol{q}_J = \boldsymbol{P}_J^T \boldsymbol{e} = \boldsymbol{0}$ is equivalent to $\boldsymbol{e} \in \mathcal{N}(\boldsymbol{C}(\mu))^\perp$.

Now suppose $\boldsymbol{e} \in \mathcal{N}(\boldsymbol{C}(\mu))^\perp$. If $\boldsymbol{C}(\mu)$ is nonsingular, then Lemma A.2 implies that $f$ is differentiable at $\mu$. If $\boldsymbol{C}(\mu)$ is singular, then $\boldsymbol{P}_J^T \boldsymbol{e} = \boldsymbol{0}$ and the two one-sided limits in the above paragraph coincide, i.e., $f$ is differentiable at $\mu$.

Equation (A.1) is a consequence of the coincidence of the one-sided limits, that the common limit does not depend on the order of $\lambda_1, \ldots, \lambda_n$, and the definition of $\phi^{[1]}$ in equation (18). $\qquad\square$

## A.2 Proof of Theorem 2

For a solution $\mu^\star$ to the equation $f(\mu) = 0$, define a collection of matrices related to the eigenvalues $\boldsymbol{\lambda}^\star = (\lambda_1^\star, \ldots, \lambda_n^\star)^T$ of $\boldsymbol{C}(\mu^\star)$:

$$\mathcal{M} = \{\boldsymbol{M} = (m_{ij}) \in \mathbb{S}^n : m_{ij} = \phi^{[1]}(\boldsymbol{\lambda}^\star) \text{ if } \lambda_i^\star \neq 0 \text{ or } \lambda_j^\star \neq 0; \; m_{ij} \in [0, 1] \text{ if } \lambda_i^\star = 0 = \lambda_j^\star\}.$$

Also define the set (Bouligand subdifferential)

$$\partial_B f(\mu^\star) = \{\lim_{k \to \infty} f'(\mu_k) : \mu_k \to \mu^\star, \; \mu_k \in D_f\}$$

where $D_f$ denotes the set of points in which $f$ is differentiable, so that $\partial f(\mu^\star) = \mathbf{conv}\, \partial_B f(\mu^\star)$. The following lemma shows a representation of an element of this set in terms of $\mathcal{M}$:

**Lemma A.3.** *Suppose a spectral decomposition of $\boldsymbol{C}(\mu^\star)$ is $\boldsymbol{P}^\star \operatorname{diag}(\lambda_1^\star, \ldots, \lambda_n^\star)\boldsymbol{P}^{\star T}$ with $\boldsymbol{P}^{\star T} \boldsymbol{P}^\star = \boldsymbol{P}^\star \boldsymbol{P}^{\star T} = \boldsymbol{I}$. Then, for any $v \in \partial_B f(\mu^\star)$, there exists $\boldsymbol{M} \in \mathcal{M}$ such that*

$$v = \boldsymbol{e}^T \boldsymbol{P}^\star (\boldsymbol{M} \circ (\boldsymbol{P}^{\star T} \boldsymbol{e}\boldsymbol{e}^T \boldsymbol{P}^\star))\boldsymbol{P}^{\star T} \boldsymbol{e}.$$

*Proof.* By the definition of $\partial_B f(\mu^\star)$, there exists a sequence $\{\mu_k\}$ such that $f$ is differentiable at each $\mu_k$, $\mu_k \to \mu^\star$, and $f'(\mu_k) \to v$ as $k \to \infty$. Obviously $\mu_k \neq \mu$ for all $k$. Thus $\boldsymbol{C}(\mu_k) = \bar{\boldsymbol{X}} - \mu_k \boldsymbol{e}\boldsymbol{e}^T = \boldsymbol{C}(\mu) - (\mu_k - \mu)\boldsymbol{e}\boldsymbol{e}^T$ is a symmetric rank-1 perturbation of $\boldsymbol{C}(\mu)$. Then, by Chen et al. [2003, Lemma 3.3], Rellich and Berkowitz [1969, Thm. 1], $\boldsymbol{C}(\mu_k)$ has a spectral decomposition $\boldsymbol{P}_k \operatorname{diag}(\lambda_{k,1}, \ldots, \lambda_{k,n})\boldsymbol{P}_k^T$ such that $\boldsymbol{P}_k \to \boldsymbol{P}^\star$ as $k \to \infty$, by passing to a subsequence of

$\{\mu_k\}$ if necessary. Since $\lambda_{k,i} = (\boldsymbol{P}_k^T \boldsymbol{C}(\mu_k) \boldsymbol{P}_k)_{ii}$ and $\boldsymbol{C}(\mu)$ is continuous in $\mu$, it follows that $\lim_{k \to \infty} \lambda_{k,i} = \lambda_i$ as well, for $i = 1, \dots, n$.

By Lemma A.1,

$$f'(\mu_k) = \boldsymbol{e}^T \boldsymbol{P}_k (\phi^{[1]}(\boldsymbol{\lambda}_k) \circ (\boldsymbol{P}_k^T \boldsymbol{e} \boldsymbol{e}^T \boldsymbol{P}_k)) \boldsymbol{P}_k^T \boldsymbol{e}.$$

Let

$$K = \{i \in \{1, \dots, n\} : \lambda_i^\star \neq 0\}, \quad J = \{i \in \{1, \dots, n\} : \lambda_i^\star = 0\}$$

and $\delta = \frac{1}{2} \min_{i \in K} |\lambda_i^\star| > 0$. Then for all sufficiently large $k$, we have $\max_{i=1,\dots,n} |\lambda_{k,i} - \lambda_i^\star| \leq \delta$. If $i \in K$ or $j \in K$, then $\lambda_{k,i} \neq 0$ or $\lambda_{k,j} \neq 0$, and

$$\phi_{ij}^{[1]}(\boldsymbol{\lambda}_k) = \frac{(\lambda_{k,i})_+ + (\lambda_{k,j})_+}{|\lambda_{k,i}| + |\lambda_{k,j}|} \to \frac{(\lambda_i^\star)_+ + (\lambda_j^\star)_+}{|\lambda_i^\star| + |\lambda_j^\star|} = \phi_{ij}^{[1]}(\boldsymbol{\lambda}^\star).$$

If $i, j \in J$, then both $\lambda_{k,i}$ and $\lambda_{k,j}$ converge to 0. Since $\phi_{i,j}(\boldsymbol{\lambda}_k) \in [0, 1]$ in this case, passing to a subsequence of $\{\mu_k\}$ if necessary, $\phi_{i,j}(\boldsymbol{\lambda}_k)$ converges to a point $m_{ij} \in [0, 1]$. This shows that $\phi^{[1]}(\boldsymbol{\lambda}_k) \to \boldsymbol{M} \in \mathcal{M}$.

Finally, by the continuity of matrix multiplications, we have

$$v = \lim_{k \to \infty} f'(\mu_k) = \boldsymbol{e}^T \boldsymbol{P}^\star (\boldsymbol{M} \circ (\boldsymbol{P}^{\star T} \boldsymbol{e} \boldsymbol{e}^T \boldsymbol{P}^\star)) \boldsymbol{P}^{\star T} \boldsymbol{e}.$$

$\square$

The next lemma provides a technical result useful for proving Theorem 2.

**Lemma A.4.** *For* $\boldsymbol{P}^\star = (p_{ij})$ *and* $\lambda_1^\star, \dots, \lambda_n^\star$ *in the statement of Lemma A.3, let* $K_+ = \{i \in \{1, \dots, n\} : \lambda_i^\star > 0\}$. *Then* $K_+ \neq \emptyset$ *and*

$$\sum_{i \in K_+} p_{ni}^2 > 0.$$

*Proof.* Denote the $i$th column of $\boldsymbol{P}$ by $\boldsymbol{p}_i = (p_{1i}, \dots, p_{ni})^T$. Then $\phi^{\square}(\boldsymbol{C}(\mu^\star)) = [\boldsymbol{C}(\mu^\star)]_+ = \sum_{i \in K_+} \lambda_i^\star \boldsymbol{p}_i \boldsymbol{p}_i^T$. From the optimality condition

$$1 = \boldsymbol{e}^T \phi^{\square}(\boldsymbol{C}(\mu^\star)) \boldsymbol{e} = \sum_{i \in K_+} \lambda_i^\star p_{ni}^2.$$

If $K_+ = \emptyset$ then the rightmost hand side is zero, a contradiction. That $K_+ \neq \emptyset$ and $\lambda_i^\star > 0$ for all $i \in K_+$ succumbs to the fact $\sum_{i \in K_+} p_{ni}^2 > 0$. $\square$

Now we are ready to prove the theorem.

*Proof of Theorem 2.* Let $v \in \partial f_B(\mu^\star)$. Also let $J$, $K$, and $K_+$ be as defined in the proof of Lemma A.3 and the statement of Lemma A.4. Define $K_- = K \setminus K_+$. Then by Lemma A.3 there exists $\boldsymbol{M} = (m_{ij}) \in \mathbb{S}^n$ such that

$$m_{ij} = \begin{cases} 1, & \text{if } i \in K_+, j \in K_+ \cup J, \text{ or } i \in J, j \in K_+, \\ 0, & \text{if } i \in J, j \in K_-, \text{ or } i \in K_-, j \in J \cup K_-, \\ \tau_{ij} = \frac{\lambda_i^\star}{\lambda_i^\star - \lambda_j^\star} \in (0, 1), & \text{if } i \in K_+, j \in K_-, \text{ or } i \in K_-, j \in K_+, \\ \in [0, 1], & \text{if } i, j \in J. \end{cases}$$

and

$$v = \boldsymbol{e}^T \boldsymbol{P}^\star [\boldsymbol{M} \circ (\boldsymbol{P}^{\star T} \boldsymbol{e} \boldsymbol{e}^T \boldsymbol{P}^\star)] \boldsymbol{P}^{\star T} \boldsymbol{e}$$

Then,

$$
\begin{aligned}
v &= \mathbf{Tr}(\boldsymbol{e}^T \boldsymbol{P}^\star [\boldsymbol{M} \circ (\boldsymbol{P}^{\star T} \boldsymbol{e}\boldsymbol{e}^T \boldsymbol{P}^\star)] \boldsymbol{P}^{\star T} \boldsymbol{e}) \\
&= \mathbf{Tr}(\boldsymbol{P}^{\star T} \boldsymbol{e}\boldsymbol{e}^T P[\boldsymbol{M} \circ (\boldsymbol{P}^{\star T} \boldsymbol{e}\boldsymbol{e}^T \boldsymbol{P}^\star)]) \\
&= \mathbf{Tr}(\boldsymbol{Q}[\boldsymbol{M} \circ \boldsymbol{Q}]), \quad \text{where } \boldsymbol{Q} = \boldsymbol{P}^{\star T} \boldsymbol{e}\boldsymbol{e}^T \boldsymbol{P}^\star = (q_{ij}) \\
&\geq \sum_{i \in K_+} \left( \sum_{j \in K_+ \cup J} q_{ij}^2 + \sum_{j \in K_-} \tau_{ij} q_{ij}^2 \right) \quad (\text{since } m_{ij} \geq 0) \\
&\geq \left( \min_{i \in K_+, j \in K_-} \tau_{ij} \right) \sum_{i \in K_+} \sum_{j=1}^n q_{ij}^2.
\end{aligned}
$$

Since $\boldsymbol{P}^{\star T} \boldsymbol{e} = (p_{n1}, \ldots, p_{nn})^T$ is the last row of $\boldsymbol{P}^\star$, we have $q_{ij} = p_{ni} p_{nj}$ and

$$
\sum_{i \in K_+} \sum_{j=1}^n q_{ij}^2 = \sum_{i \in K_+} \sum_{j=1}^n p_{ni}^2 p_{nj}^2 = \left( \sum_{i \in K_+} p_{ni}^2 \right) \left( \sum_{j=1}^n p_{nj}^2 \right) > 0.
$$

The quantity is the first pair of parentheses is positive due to Lemma A.4. The second quantity equals to $\boldsymbol{e}^T \boldsymbol{P}^\star \boldsymbol{P}^{\star T} \boldsymbol{e} = \boldsymbol{e}^T \boldsymbol{e} = 1$. From this and $\tau_{ij} > 0$ for all $i \in K_+$ and $j \in K_-$, it follows that $v > 0$.

Since $\partial_B f(\mu^\star)$ is compact and all the elements of this set is positive, and convex combination of its elements is also positve. It follows that every element of $\partial f(\mu^\star) = \mathbf{conv}\, \partial_B f(\mu^\star)$ is positive.

The uniqueness of solution then follows from Clarke's inverse function theorem [Clarke, 1990, Thm. 7.1.1]; existence of solution is shown in Section 2 of the main text. □

### A.3  Proof of Theorem 4

The proof of Theorem 4 also requires Lemma A.1.

*Proof of Theorem 4.* If $f$ is differentiable at $\mu$, then $\partial f(\mu) = \{f'(\mu)\}$ and the result holds by Lemma A.1. Otherwise, consider a sequence $\{\mu_k\}$ such that $\mu_k \downarrow \mu$ and $f$ is differentiable at each $\mu_k$. Such a sequence exists since $f$ is Lipschitz hence almost everywhere differentiable [Rockafellar and Wets, 2009, sec. 9J]. Obviously $\mu_k > \mu$ for all $k$. Thus $\boldsymbol{C}(\mu_k) = \bar{\boldsymbol{X}} - \mu_k \boldsymbol{e}\boldsymbol{e}^T = \boldsymbol{C}(\mu) - (\mu_k - \mu)\boldsymbol{e}\boldsymbol{e}^T$ is a symmetric rank-1 perturbation of $\boldsymbol{C}(\mu)$. Then, by Chen et al. [2003, Lemma 3.3], Rellich and Berkowitz [1969, Thm. 1], $\boldsymbol{C}(\mu_k)$ has a spectral decomposition $\boldsymbol{P}_k \operatorname{diag}(\lambda_{k,1}, \ldots, \lambda_{k,n}) \boldsymbol{P}_k^T$ such that $\boldsymbol{P}_k \to \boldsymbol{P}$ as $k \to \infty$, by passing to a subsequence if necessary. Since $\lambda_{k,i} = (\boldsymbol{P}_k^T \boldsymbol{C}(\mu_k) \boldsymbol{P}_k)_{ii}$ and $\boldsymbol{C}(\mu)$ is continuous in $\mu$, it follows that $\lim_{k \to \infty} \lambda_{k,i} = \lambda_i$ as well, for $i = 1, \ldots, n$. Moreover, $\lambda_{k,i} \leq \lambda_i$ for all $i$ [Bunch et al., 1978]. Thus if $\lambda_i = 0 = \lambda_j$, then $\lambda_{k,i}, \lambda_{k,j} \uparrow 0$, which implies that $\lim_{k \to \infty} \phi^{[1]}(\boldsymbol{\lambda}_k) = \phi^{[1]}(\boldsymbol{\lambda})$. Now since from Lemma A.1,

$$
f'(\mu_k) = \boldsymbol{e}^T \boldsymbol{P}_k (\phi^{[1]}(\boldsymbol{\lambda}_k) \circ (\boldsymbol{P}_k^T \boldsymbol{e}\boldsymbol{e}^T \boldsymbol{P}_k)) \boldsymbol{P}_k^T \boldsymbol{e}, \quad \boldsymbol{\lambda}_k = (\lambda_{k,1}, \ldots, \lambda_{k,n})^T,
$$

it follows that $\lim_{k \to \infty} f'(\mu_k) = v$. From Definition 1, we see $v \in \partial f(\mu)$. □

## B  Applications to proximal algorithms

### B.1  Heteroskedastic scaled lasso

In the heteroskedastic scaled lasso we want to minimize

$$
\ell(\boldsymbol{\Omega}, \boldsymbol{\beta}) = \phi(\boldsymbol{\Omega}, \boldsymbol{X}\boldsymbol{\beta} - \boldsymbol{y}) + \frac{1}{2\sqrt{N}} \|\boldsymbol{\Omega}\|_F + \lambda \|\boldsymbol{\beta}\|_1. \tag{B.1}
$$

If we define the affine map $\mathcal{K} : (\boldsymbol{\Omega}, \boldsymbol{\beta}) \mapsto (\boldsymbol{\Omega}, \boldsymbol{X}\boldsymbol{\beta} - \boldsymbol{y})$, then problem (B.1) has the form (5), where $f(\boldsymbol{\Omega}, \boldsymbol{\beta}) \equiv 0$, $g(\boldsymbol{\Omega}, \boldsymbol{\beta}) = \frac{1}{2\sqrt{N}} \|\boldsymbol{\Omega}\|_F + \lambda \|\boldsymbol{\beta}\|_1$, and $h = \phi$. The adjoint $\mathcal{K}^T$ of the linear part of $\mathcal{K}$

maps $(\mathbf{\Theta}, \boldsymbol{\zeta}) \in \mathbb{S}^p \times \mathbb{R}^p$ to $(\mathbf{\Theta}, \boldsymbol{X}^T \boldsymbol{\zeta})$. Thus the resulting PDHG iteration is

$$\mathbf{\Omega}^{k+1} = \left(1 - \frac{\tau/(2\sqrt{N})}{\max[\|\boldsymbol{Y}\|_F, \tau/(2\sqrt{N})]}\right) \boldsymbol{Y}, \quad \boldsymbol{Y} = \mathbf{\Omega}^k - \tau\mathbf{\Theta}^k,$$

$$\boldsymbol{\beta}^{k+1} = S_{\tau\lambda}\left(\boldsymbol{\beta}^k - \tau\boldsymbol{X}^T\boldsymbol{\zeta}^k\right),$$

$$\tilde{\mathbf{\Omega}}^{k+1} = 2\mathbf{\Omega}^{k+1} - \mathbf{\Omega}^k,$$

$$\tilde{\boldsymbol{\beta}}^{k+1} = 2\boldsymbol{\beta}^{k+1} - \boldsymbol{\beta}^k,$$

$$(\mathbf{\Theta}^{k+1}, \boldsymbol{\zeta}^{k+1}) = \mathbf{prox}_{\sigma\phi^*}\left(\mathbf{\Theta}^k + \sigma\tilde{\mathbf{\Omega}}^{k+1}, \; \boldsymbol{\zeta}^k + \sigma(\boldsymbol{X}\tilde{\boldsymbol{\beta}}^{k+1} - \boldsymbol{y})\right).$$

where $S_{\tau\lambda}$ is the usual soft-thresholding operator: $[S_{\tau\lambda}(\boldsymbol{x})]_i = \min(\max(x_i - \tau\lambda, 0), x_i + \tau\lambda)$.

In order to determine the step sizes, note $\mathcal{K}^T\mathcal{K} : (\mathbf{\Omega}, \beta) \mapsto (\mathbf{\Omega}, \boldsymbol{X}^T\boldsymbol{X}\boldsymbol{\beta} - \boldsymbol{X}^T\boldsymbol{y})$. The norm of the linear part of this affine operator equals $\max(\|\boldsymbol{X}^T\boldsymbol{X}\|_2, 1) = \max(\|\boldsymbol{X}\|_2^2, 1) \leq \max(\|\boldsymbol{X}\|_F^2, 1)$.

**Setup for experiments** For all combinations of $(N, p)$ in Table 2, data matrix $\boldsymbol{X} \in \mathbb{R}^{N \times p}$ were generated from zero-mean independent Gaussian. Each $\boldsymbol{x}_i$ was then scaled to have norm $1/\sqrt{p}$, so that $\|\boldsymbol{X}\|_F = 1$. Response vector $\boldsymbol{y}$ was generated by setting $\boldsymbol{y} = \boldsymbol{X}\boldsymbol{\beta} + \boldsymbol{\epsilon}$, where the first five components of $\boldsymbol{\beta}$ were independently generated from $\mathcal{N}(0, 10^2)$ and the rest set to zero; noise vector $\boldsymbol{\epsilon}$ was generated from zero-mean $n$-variate Gaussian with covariance matrix of compound symmetry

$$\mathbf{\Sigma} = \begin{bmatrix} 1 & \rho & \rho & \cdots & \rho \\ \rho & 1 & \rho & \cdots & \rho \\ \vdots & & \ddots & & \vdots \\ \rho & \rho & \rho & \cdots & 1 \end{bmatrix}$$

with $\rho = 0.5$. The regularization parameter $\lambda = 0.005$. The PDHG iteration was initialized by $\mathbf{\Omega}^0 = \boldsymbol{I}_N$, $\boldsymbol{\beta}^0 = \mathbf{0}$, $\mathbf{\Theta}^0 = \mathbf{0}$, and $\boldsymbol{\zeta}^0 = \mathbf{0}$. The step size parameters are $\tau = 0.99$ and $\sigma = 0.99$. Convergence was declared when the relative change of the primal variables $(\mathbf{\Omega}^k, \boldsymbol{\beta}^k)$ was less than $10^{-6}$ for $p < 300$ and $10^{-5}$ for $p \geq 300$. The maximum number of iterations was set to 50000.

## B.2 Gaussian joint likelihood estimation

Joint maximum likelihood estimation (MLE) of Gaussian natural parameters $(\mathbf{\Omega}, \boldsymbol{\eta})$ under the variance constraints

$$\begin{aligned} \text{minimize} \quad & \ell(\mathbf{\Omega}, \boldsymbol{\eta}) = -\log\det\mathbf{\Omega} + \mathbf{Tr}(\mathbf{\Omega}\boldsymbol{S}) - 2\bar{\boldsymbol{\mu}}^T\boldsymbol{\eta} + \phi(\mathbf{\Omega}, \boldsymbol{\eta}) + \frac{\epsilon}{2}\|\mathbf{\Omega}\|_F^2 \\ \text{subject to} \quad & \boldsymbol{c}_i^T\mathbf{\Omega}^{-1}\boldsymbol{c}_i \leq 1, \quad i = 1, \ldots, m \end{aligned} \tag{B.2}$$

(the ridge penalty $\frac{\epsilon}{2}\|\mathbf{\Omega}\|_F^2$ is added to ensure existence of the solution) has the form (5) if we define

$$f(\mathbf{\Omega}, \boldsymbol{\eta}) = 0$$

$$g(\mathbf{\Omega}, \boldsymbol{\eta}) = -\log\det\mathbf{\Omega} + \mathbf{Tr}(\mathbf{\Omega}\boldsymbol{S}) - 2\bar{\boldsymbol{\mu}}^T\boldsymbol{\eta} + \frac{\epsilon}{2}\|\mathbf{\Omega}\|_F^2$$

$$h(\boldsymbol{Z}_0, \boldsymbol{Z}_1, \cdots, \boldsymbol{Z}_m, \boldsymbol{\eta}) = \phi(\boldsymbol{Z}_0, \boldsymbol{\eta}) + \sum_{i=1}^m \iota_{C_i}(\boldsymbol{Z}_i), \quad C_i = \{\mathbf{\Omega} \in \mathbb{S}^p : \boldsymbol{c}_i^T\mathbf{\Omega}^{-1}\boldsymbol{c}_i \leq 1\},$$

and the linear map $\mathcal{K} : (\mathbf{\Omega}, \boldsymbol{\eta}) \mapsto (\mathbf{\Omega}, \mathbf{\Omega}, \ldots, \mathbf{\Omega}, \boldsymbol{\eta}) \in \prod_{i=0}^m \mathbb{S}^p \times \mathbb{R}^p$.

Since the adjoint $\mathcal{K}^T$ of $\mathcal{K}$ maps $(\boldsymbol{\Theta}_0, \boldsymbol{\Theta}_1, \ldots, \boldsymbol{\Theta}_m, \boldsymbol{\zeta}) \in \prod_{i=0}^m \mathbb{S}^p \times \mathbb{R}^p$ to $(\sum_{i=0}^m \boldsymbol{\Theta}_i, \boldsymbol{\zeta})$, the PDHG iteration for problem (B.2) entails

$$\boldsymbol{\Omega}^{k+1} = \mathbf{prox}_{-\frac{\tau}{1+\epsilon\tau} \log \det(\cdot)} \left( \frac{1}{1+\epsilon\tau}(\boldsymbol{\Omega}^k - \tau \sum_{i=0}^m \boldsymbol{\Theta}_i^k - \tau \boldsymbol{S}) \right),$$

$$\boldsymbol{\eta}^{k+1} = \boldsymbol{\eta}^k - \tau \boldsymbol{\zeta}^k + 2\tau \bar{\boldsymbol{\mu}},$$

$$\tilde{\boldsymbol{\Omega}}^{k+1} = 2\boldsymbol{\Omega}^{k+1} - \boldsymbol{\Omega}^k,$$

$$\tilde{\boldsymbol{\eta}}^{k+1} = 2\boldsymbol{\eta}^{k+1} - \boldsymbol{\eta}^k,$$

$$(\boldsymbol{\Theta}_0^{k+1}, \boldsymbol{\zeta}^{k+1}) = \mathbf{prox}_{\sigma \phi^*} \left( \boldsymbol{\Theta}_0^k + \sigma \tilde{\boldsymbol{\Omega}}^{k+1}, \boldsymbol{\zeta}^k + \sigma \tilde{\boldsymbol{\eta}}^{k+1} \right),$$

$$\boldsymbol{\Theta}_i^{k+1} = \mathbf{prox}_{\sigma \iota_{C_i}^*} \left( \boldsymbol{\Theta}_i^k + \sigma \tilde{\boldsymbol{\Omega}}^{k+1} \right), \quad i = 1, \ldots, m.$$

It is well-known that

$$\mathbf{prox}_{-\tau \log \det(\cdot)}(\boldsymbol{M}) = \boldsymbol{Q} \operatorname{diag} \left( \frac{\mu_1 + \sqrt{\mu_1^2 + 4\tau}}{2}, \ldots, \frac{\mu_p + \sqrt{\mu_p^2 + 4\tau}}{2} \right) \boldsymbol{Q}^T$$

if the eigenvalue decomposition of $\boldsymbol{M} \in \mathbb{S}^p$ is $\boldsymbol{Q} \operatorname{diag}(\mu_1, \ldots, \mu_p)\boldsymbol{Q}^T$.

It remains to compute $\mathbf{prox}_{\sigma \iota_{C_i}^*}$. The following result shows it has a closed-form expression.

**Proposition B.1.** *Let $S_{\boldsymbol{c}, \alpha} = \{\boldsymbol{\Omega} \in \mathbb{S}^p : \phi(\boldsymbol{\Omega}, \boldsymbol{c}) \leq \alpha\}$ where $\alpha > 0$. Then $S_{\boldsymbol{c}, \alpha}$ is closed and convex. Furthermore, the projection of $\boldsymbol{Z} \in \mathbb{S}^p$ onto $S_{\boldsymbol{c}, \alpha}$ is*

$$P_{S_{\boldsymbol{c}, \alpha}}(\boldsymbol{Z}) = \left( \boldsymbol{Z} - \frac{1}{2\alpha}\boldsymbol{c}\boldsymbol{c}^T \right)_+ + \frac{1}{2\alpha}\boldsymbol{c}\boldsymbol{c}^T.$$

Therefore, from the Moreau decomposition (7), for $i = 1, \ldots, m$,

$$\mathbf{prox}_{\sigma \iota_{C_i}^*}(\boldsymbol{Y}) = \boldsymbol{Y} - \sigma P_{S_{\boldsymbol{c}_i, 1/2}}(\sigma^{-1}\boldsymbol{Y}) = \sigma \left( \frac{1}{\sigma}\boldsymbol{Y} - \boldsymbol{c}_i \boldsymbol{c}_i^T \right) - \sigma \left( \frac{1}{\sigma}\boldsymbol{Y} - \boldsymbol{c}_i \boldsymbol{c}_i^T \right)_+$$

$$= -\sigma \left( \boldsymbol{c}_i \boldsymbol{c}_i^T - \frac{1}{\sigma}\boldsymbol{Y} \right)_+.$$

Finally, to determine the step sizes, note $\mathcal{K}^T \mathcal{K} : (\boldsymbol{\Omega}, \boldsymbol{\eta}) \mapsto ((m+1)\boldsymbol{\Omega}, \boldsymbol{\eta})$. Hence $\|\mathcal{K}^T \mathcal{K}\|_2 = m+1$.

*Proof of Proposition B.1.* Convexity and closedness of $S_{\boldsymbol{c}, \alpha}$ follows from those of $\phi$. The projection operator is

$$P_{S_{\boldsymbol{c}, \alpha}}(\boldsymbol{Z}) = \operatorname*{arg\,min}_{\boldsymbol{\Omega} \in \mathbb{S}^p} \frac{1}{2}\|\boldsymbol{Z} - \boldsymbol{\Omega}\|_F^2 \text{ subject to } \phi(\boldsymbol{\Omega}, \boldsymbol{c}) \leq \alpha$$

$$= \operatorname*{arg\,min}_{\boldsymbol{\Omega} \in \mathbb{S}^p} \frac{1}{2}\|\boldsymbol{Z} - \boldsymbol{\Omega}\|_F^2 \text{ subject to } \frac{1}{2}\boldsymbol{c}^T \boldsymbol{\Omega}^\dagger \boldsymbol{c} \leq \alpha, \ \boldsymbol{c} \in \mathcal{R}(\boldsymbol{\Omega})$$

$$= \operatorname*{arg\,min}_{\boldsymbol{\Omega} \in \mathbb{S}^p} \frac{1}{2}\|\boldsymbol{Z} - \boldsymbol{\Omega}\|_F^2 \text{ subject to } \alpha - \frac{1}{2}\boldsymbol{c}^T \boldsymbol{\Omega}^\dagger \boldsymbol{c} \geq 0, \ \boldsymbol{c} \in \mathcal{R}(\boldsymbol{\Omega})$$

$$= \operatorname*{arg\,min}_{\boldsymbol{\Omega} \in \mathbb{S}^p} \frac{1}{2}\|\boldsymbol{Z} - \boldsymbol{\Omega}\|_F^2 \text{ subject to } \boldsymbol{\Omega} - \frac{1}{2\alpha}\boldsymbol{c}\boldsymbol{c}^T \succeq \boldsymbol{0}$$

$$= \operatorname*{arg\,min}_{\boldsymbol{\Omega} \in \mathbb{S}^p} \frac{1}{2}\left\| \boldsymbol{Z} - \frac{1}{2\alpha}\boldsymbol{c}\boldsymbol{c}^T - \left( \boldsymbol{\Omega} - \frac{1}{2\alpha}\boldsymbol{c}\boldsymbol{c}^T \right) \right\|_F^2 \text{ subject to } \boldsymbol{\Omega} - \frac{1}{2\alpha}\boldsymbol{c}\boldsymbol{c}^T \succeq \boldsymbol{0}$$

$$= \left( \boldsymbol{Z} - \frac{1}{2\alpha}\boldsymbol{c}\boldsymbol{c}^T \right)_+ + \frac{1}{2\alpha}\boldsymbol{c}\boldsymbol{c}^T.$$

The fourth equality is due to the Schur complements of

$$\begin{bmatrix} \boldsymbol{\Omega} & -\frac{1}{\sqrt{2}}\boldsymbol{c} \\ -\frac{1}{\sqrt{2}}\boldsymbol{c}^T & \alpha \end{bmatrix} \succeq \boldsymbol{0}.$$

The last equality is from the fact $\operatorname*{arg\,min}_{\boldsymbol{X} \succeq \boldsymbol{0}} \frac{1}{2}\|\boldsymbol{Z} - \boldsymbol{X}\|_F^2 = \boldsymbol{Z}_+$. $\qquad \square$

**Setup for experiments**  For all combinations of $(N, p)$ in Table 2, data $\boldsymbol{x}_1, \ldots, \boldsymbol{x}_N \in \mathbb{R}^p$ were generated from zero-mean multivariate Gaussian with covariance matrix of compound symmetry

$$\boldsymbol{\Sigma} = \begin{bmatrix} 1 & \rho & \rho & \cdots & \rho \\ \rho & 1 & \rho & \cdots & \rho \\ \vdots & & \ddots & & \vdots \\ \rho & \rho & \rho & \cdots & 1 \end{bmatrix}$$

with $\rho = 0.3$. The PDHG iteration used $\epsilon = 10/p^2$ and was initialized by

$$\boldsymbol{\Omega}^0 = (\boldsymbol{S} - \boldsymbol{\mu}\boldsymbol{\mu}^T + 10^{-2}\boldsymbol{I}_p)^{-1}$$
$$\boldsymbol{\eta}^0 = \boldsymbol{\Omega}^0\bar{\boldsymbol{\mu}}$$
$$\boldsymbol{\Theta}_i^0 = \boldsymbol{\Omega}^0, \quad i = 0, 1, \ldots, m$$
$$\boldsymbol{\zeta}^0 = \boldsymbol{\eta}^0.$$

The step size parameters are $\tau = 1$ and $\sigma = 1/(m+1)$. Convergence was declared when the relative change of the primal variables $(\boldsymbol{\Omega}^k, \boldsymbol{\eta}^k)$ was less than $10^{-5}$. The maximum number of iterations was set to 50000.

## B.3  Graphical model selection

Recall from equation (3) we want to minimize

$$-\frac{1}{N}PL(\boldsymbol{\Omega}) + \lambda|\boldsymbol{\Omega}|_1 = -\frac{1}{2}\sum_{i=1}^p \log\omega_{ii} + \phi(\mathcal{K}\boldsymbol{\Omega}) + \lambda\sum_{i<j}|\omega_{ij}|. \tag{B.3}$$

This has the form (5) if we define $f(\boldsymbol{\Omega}) \equiv 0$, $g(\boldsymbol{\Omega}) = -\frac{1}{2}\sum_{i=1}^p \log\omega_{ii} + \lambda\sum_{i<j}|\omega_{ij}|$, $h = \phi$, and the linear map $\mathcal{K} : \boldsymbol{\Omega} \mapsto \frac{1}{N}(\boldsymbol{I}_N \otimes \boldsymbol{\Omega}_D, \mathbf{vec}(\boldsymbol{\Omega}\boldsymbol{Y}^T))$. The adjoint of $\mathcal{K}$ is

$$\mathcal{K}^T : (\boldsymbol{M}, \mathbf{vec}(\boldsymbol{Z})) \mapsto \frac{1}{N}\sum_{i=1}^N \boldsymbol{M}_{ii,D} + \frac{1}{2N}(\boldsymbol{Z}\boldsymbol{Y} + \boldsymbol{Y}^T\boldsymbol{Z}^T),$$

for symmetric block matrix $\boldsymbol{M} = (\boldsymbol{M}_{ij}) \in \mathbb{S}^{Np}$ with $\boldsymbol{M}_{ij} = \boldsymbol{M}_{ji}^T \in \mathbb{R}^{p\times p}$, and $\boldsymbol{Z} \in \mathbb{R}^{p\times N}$. Then the PDHG iteration for problem (B.3) is

$$\boldsymbol{\Omega}^{k+1} = \mathbf{prox}_{\tau g}\left(\boldsymbol{\Omega}^k - \frac{\tau}{N}\left(\sum_{i=1}^N \boldsymbol{\Theta}_{ii,D}^k + \frac{1}{2}\boldsymbol{Z}^k\boldsymbol{Y} + \frac{1}{2}\boldsymbol{Y}^T[\boldsymbol{Z}^k]^T\right)\right)$$

$$\tilde{\boldsymbol{\Omega}}^{k+1} = 2\boldsymbol{\Omega}^{k+1} - \boldsymbol{\Omega}^k$$

$$(\boldsymbol{\Theta}^{k+1}, \mathbf{vec}(\boldsymbol{Z}^{k+1})) = \mathbf{prox}_{\sigma\phi^*}\left(\boldsymbol{\Theta}^k + \frac{\sigma}{N}(\boldsymbol{I}_N \otimes \tilde{\boldsymbol{\Omega}}_D^{k+1}), \mathbf{vec}(\boldsymbol{Z}^k + \frac{\sigma}{N}\tilde{\boldsymbol{\Omega}}^{k+1}\boldsymbol{Y}^T)\right)$$

where $\boldsymbol{\Omega}^k, \tilde{\boldsymbol{\Omega}}^k \in \mathbb{S}^p$, $\boldsymbol{Z}^k \in \mathbb{R}^{p\times N}$, and $\boldsymbol{\Theta}^k = (\boldsymbol{\Theta}_{ij}^k) \in \mathbb{S}^{Np}$, with $\boldsymbol{\Theta}_{ij} = \boldsymbol{\Theta}_{ji}^T \in \mathbb{R}^{p\times p}$. Operator $\mathbf{prox}_{\tau g}$ has a closed form expression. For $\boldsymbol{W} = (w_{ij})$,

$$[\mathbf{prox}_{\tau g}(\boldsymbol{W})]_{ij} = \begin{cases} \frac{1}{2}(w_{ii} + \sqrt{w_{ii}^2 + 2\tau}), & i = j, \\ S_{\tau\lambda/2}(w_{ij}), & i \neq j. \end{cases}$$

It is easy to see that $\mathcal{K}^T\mathcal{K} : \boldsymbol{\Omega} \mapsto \frac{1}{N}\boldsymbol{\Omega}_D + \frac{1}{2N^2}(\boldsymbol{\Omega}\boldsymbol{Y}^T\boldsymbol{Y} + \boldsymbol{Y}^T\boldsymbol{Y}\boldsymbol{\Omega})$. Then $\mathbf{vec}(\frac{1}{N}\boldsymbol{\Omega}_D + \frac{1}{2N^2}[\boldsymbol{\Omega}\boldsymbol{Y}^T\boldsymbol{Y} + \boldsymbol{\Omega}\boldsymbol{Y}^T\boldsymbol{Y}]) = \left(\frac{1}{N}\boldsymbol{A} + \frac{1}{2N^2}(\boldsymbol{Y}^T\boldsymbol{Y} \otimes \boldsymbol{I}_p + \boldsymbol{I}_p \otimes \boldsymbol{Y}^T\boldsymbol{Y})\right)\mathbf{vec}(\boldsymbol{\Omega})$ where $\boldsymbol{A}$ satis-

fies $\textbf{vec}(\boldsymbol{\Omega}_D) = \boldsymbol{A}\,\textbf{vec}(\boldsymbol{\Omega})$. It follows that $\boldsymbol{A}^T\boldsymbol{A} = \boldsymbol{I}_{p^2}$ and $\|\boldsymbol{A}\|_2 = 1$. Therefore,

$$
\begin{aligned}
\|\mathcal{K}^T\mathcal{K}\|_2 &= \left\| \frac{1}{N}\boldsymbol{A} + \frac{1}{2N^2}(\boldsymbol{Y}^T\boldsymbol{Y}\otimes\boldsymbol{I}_p + \boldsymbol{I}_p\otimes\boldsymbol{Y}\boldsymbol{Y}^T) \right\|_2 \\
&\leq \frac{1}{N}\|\boldsymbol{A}\|_2 + \frac{1}{2N^2}\|\boldsymbol{Y}^T\boldsymbol{Y}\otimes\boldsymbol{I}_p\|_2 + \frac{1}{2N^2}\|\boldsymbol{I}_p\otimes\boldsymbol{Y}^T\boldsymbol{Y}\|_2 \\
&= \frac{1}{N}(1) + \frac{1}{2N^2}\lambda_{\max}(\boldsymbol{Y}^T\boldsymbol{Y}) + \frac{1}{2n^2}\lambda_{\max}(\boldsymbol{Y}^T\boldsymbol{Y}) \\
&= \frac{1}{N} + \frac{1}{N^2}\|\boldsymbol{Y}\|_2^2 \\
&\leq \frac{1}{N} + \frac{1}{N^2}\|\boldsymbol{Y}\|_F^2,
\end{aligned}
$$

which determines the step size.

**Setup for experiments**  For all combinations of $(N, p)$ in Table 2, data $\boldsymbol{y}_1, \ldots, \boldsymbol{y}_N \in \mathbb{R}^p$ were generated from zero-mean multivariate Gaussian with precision matrix

$$
\boldsymbol{\Omega} = 10\boldsymbol{I}_p + \boldsymbol{\Xi} + \boldsymbol{\Xi}^T,
$$

where $\boldsymbol{\Xi}$ is a $p \times p$ sparse random Gaussian matrix with 1 percent sparsity level. The regularization parameter $\lambda = 0.1$. The PDHG iteration was initialized by

$$
\begin{aligned}
\boldsymbol{\Omega}^0 &= (\boldsymbol{S} + 10^{-2}\boldsymbol{I}_p)^{-1} \\
\boldsymbol{\Theta}_i^0 &= \boldsymbol{I}_N \otimes \boldsymbol{\Omega}_D^0 \\
\boldsymbol{Z}^0 &= \boldsymbol{\Omega}^0\boldsymbol{Y}^T.
\end{aligned}
$$

The step size parameters are $\tau = 2$ and $\sigma = 1/(2L_K)$ where $L_K = 1/N + \|\boldsymbol{Y}\|_F^2/N^2$. Convergence was declared when the relative change of the primal variable $\boldsymbol{\Omega}^k$ was less than $10^{-5}$. The maximum number of iterations was set to 50000. For the symmetric lasso used for comparison the implementation in the gconcord R package (https://cran.r-project.org/web/packages/gconcord/index.html) was used with the same input.