[Reviews · NeurIPS 2020]

Review 1

Summary and Contributions: ### Update ### I have read the rebuttal and the other reviews. I'd like to thank to the authors for their carefully thought-out response. In general, I agree with their position that the organization and exposition are the weakest aspects of the submission. I have increased my score to 7 given the authors' commitment to improve the text and address my and the other reviewers' specific comments. Some point-by-point comments follow: 1) Great, I think readability will be greatly improved when the two sections are merged. 2) - v_k \in \ [0,1]: thanks for addressing this and updating the manuscript accordingly. - Convergence of v_k: my understanding is that the Bolzano-Weierstrauss theorem only guarantees that a sub-sequence converges. Unless v_k is monotone, I still can't see how the overall sequence can be shown to converge. 3) Yes, I agree the result is global. Thanks for the correction. 4) I didn't mean to say that analysis from Boyd and Vandenberghe could be replicated in this case — only that such analyses are more satisfying. ============ This submission proposes a guarded Newton method for evaluating the proximal operator of the matrix perspective function. The authors show that the proximal operator can be evaluated in "almost closed-form" as the solution to a 1-dimensional, non-smooth root-finding problem. This root-finding problem is characterized and shown to have a unique solution. The authors also re-derive the proximal operator as the support function of a convex cone C, which uncovers a connection between projections onto C and evaluating the prox. Inspired by this connection, a Newton-type method for solving the root-finding problem is described and proved to obtain an quadratic (asymptotic) convergence rate. Several example machine learning problems where the matrix perspective function arises are detailed. An empirical evaluation of the proposed Newton method is conducted for synthetic versions of the following problems: the lasso with heteroskedastic noise, Gaussian maximum likelihood with covariance constraints, and pseudo-likelihood maximization in a graphical model. The new algorithm is shown to be far more efficient and scalable than commercial software which naively computes the prox as a semi-definite program.

Strengths: The main strengths of this submission are (a) the inventive and interesting analysis surrounding the matrix perspective function and (b) the impressive practical performance of the guarded Newton method. (a) The analysis, particularly on the connection between the matrix perspective function and matrix nearness problem, is cleverly developed and showcases a variety of useful tricks in matrix and non-smooth analysis. For example, tools for matrix inequality constraints, variational representations of matrix functions, and computing generalized subgradients are used frequently. Overall, I enjoyed reading this work and learned a fair amount. (b) Empirical evaluations show that guarded Newton is much faster and more scalable than the commercial software MOSEK, which directly solves the proximal operator as a semi-definite program. Indeed, MOSEK does not scale beyond instances of dimension 100, while Newton scales to $d = 2000$ and is still faster than MOSEK with $d = 100$. Scalability is critical for modern, high-dimensional machine learning problems and so this advance appears quite meaningful to me.

Weaknesses: This submission is highly technical and highly specific. The text is dense and no attempt is made to make the mathematical content approachable for a wider audience. I think that only readers already knowledgeable in convex analysis will benefit from reading the manuscript in it's current form. While there are undoubtedly many people in the NeurIPS community who will enjoy and benefit from this in-depth discussion of the matrix perspective function, it is important to recognize that it is inaccessible to the broader community. I think that some of the technical discussion is unnecessary. Specifically, Section 2, which provides one derivation for the proximal operator of the matrix perspective function, serves little purpose since Section 3 provides a more straightforward (no free parameter $c$ or constraint on $\mu$) derivation of an identical result. My guess is that the authors discovered the derivations in this order. Unfortunately, Section 2 comes across more as filler to pad the paper than as useful, additional discussion. This is especially true given that the derivation in Section 3 is also the one which exposes the connection to the matrix nearness problem. The convergence result for the guarded Newton method is asymptotic and local (it only holds after a sufficient number of iterations have passed). A global, non-asymptotic analysis is much more compelling for machine learning problems, where only a finite number of iterations will ever be used. An alternative and more satisfying analysis framework is two-phase convergence, such as can be found in Boyd and Vadenberghe ([2] in the manuscript). Finally, a minor weakness it that the experiments are only for synthetic problems. Real-data experiments more compelling, although I still found the empirical evaluation highly convincing. ---- Summary ---- Overall, I think that the submission is borderline. The analysis is correct and insightful and the proposed algorithm appears to be a large improvement over existing methods. On the other-hand, the work is narrow in scope, short on content, and is aimed at a relatively small subset of the NeurIPS community.

Correctness: I checked the derivations in the submission and supplement. They appear to be correct. I have several minor comments and questions as follows: - Line 25: I believe $t \Omega$ should be $t I_p$? - Proof of Theorem 1: The "Y" argument to the proximal operator shouldn't be capitalized. - The notation in Eq. (9) is a bit hard to follow, since the arguments of the argmin, $\Lambda$ and $\mu$, are not the solutions to the proximal operator; rather, the sub-matrices of $\Lambda$ are the solution. - It would be nice if several more lines of derivation were added to Section 3 in order to show that the solution to Eq. (16) leads to an equivalent expression for the prox to the one derived in Section 2. Checking this is fairly non-trivial -- it requires deriving the expression for $U$ in terms of $\Lambda$ and $\mu$, solving Eq. (16), and then substituting into the equation just before Line 94. - Line 151: Why do we need k sufficiently large for $0 \leq v_k \leq 1$? Shouldn't this immediately hold for all $k$ by monotonicity and 1-Lipschitzness of f? - Line 151: Why does the existence of a limit point for $v_k$ follow the fact that $v_k \in [0,1]$ for sufficiently large k?

Clarity: As mentioned in the weaknesses section, the paper is quite dense and can be hard to read at times. The proof of Theorem 3 is especially difficult to follow due to the amount of inline math and would benefit if some expressions were instead typeset in display mode. Sometimes additional symbols with identical meanings to existing notation are introduced (e.g. $\g'(\mu)$ and $f(\mu)$), which can confuse the reader. I suggest these symbols be merged. I strongly recommend that the authors try to reduce the technicality of the discussion. This is especially important for sections like the introduction, which should appeal to a broader audience.

Relation to Prior Work: I am unfamiliar with the literature in this area, but relevant prior work is appears to property cited. With that said, I think the manuscript would strongly benefit from a focused discussion of existing algorithms for evaluating the proximal operator. For example, is evaluating the prox as a semi-definite program (e.g. using MOSEK) the main alternative to guarded Newton, or do additional approaches exist? More context for the analysis and algorithm proposed in the submission will clarify the novelty of the research, especially for readers who are not specialists in this area.

Reproducibility: Yes

Additional Feedback: I'd like to thank the authors for an interesting submission. I enjoyed reading it and learned a few nice tools for matrix analysis.


Review 2

Summary and Contributions: ****** Post-Rebuttal ******* Overall I was fairly positive on this paper from my initial review, and most of my concerns in the original review were largely issues with writing and motivation of certain aspects of the approach, which I think should be fairly easy for the authors to address in a final version. ********************** In this paper the authors derive a means to efficiently calculate the proximal operator of the matrix projection function, which arises in applications of Gaussian likelihood estimation as well as several other problems. The main contribution is to show that the proximal operator can be found by solving a 1-dimensional (once differentiable) convex optimization problem followed by a closed-form calculation – the Euclidean projection of a matrix onto the set of negative semi-definite matrices. The authors additionally provide a Newton algorithm to solve the 1-dimensional optimization problem with quadratic convergence and point out connections between the problem of interest and the matrix nearness problem. Experiments are also provided to show the efficiency/scalability of the derived method compared to solving a SDP form of the problem with a standard SDP solver (MOSEK).

Strengths: Overall I found the paper to be technically sound. I read the proof of the main result (Theorem 1 + other derivations in sections 2-3) in some detail and am convinced of its correctness. The remaining results also appear correct, although I did not read them as carefully. The proposed approach clearly has significant advantages over the naive approach of attempting to solve the problem as a SDP. The proposed approach is also clearly scalable and straight-forward to apply directly to any problem size for which one can feasibly compute the SVD of the input matrix.

Weaknesses: I would largely consider most of the weaknesses to be issues with motivation and presentation rather than with the technical content of the results. 1) The motivation/need for the Newton algorithm in section 4 was somewhat lacking I felt. This is essentially just a 1-dimensional line search on a convex function, so even something as basic as a bisecting line search will converge linearly. While of course quadratic convergence is better than linear convergence, how much of an impact does this actually make on the run-time of the algorithm? Experiments along these lines would help motivate the need for the analysis/algorithm. 2) The introduction would benefit from some simple organization. As written all of the applications on page 2 are somewhat mashed together without clear transitions between topics. Simply making a subsection like “Applications of the Matrix Perspective Function”, then having \paragraph{Gaussian Likelihood Estimation}, \paragraph{Graphical Model Selection}, etc would significantly improve the readability of the introduction in my view. 3) At a high level there is the question of whether an entire paper devoted to computing a proximal operator is warranted (as this is typically an intermediate result given in a paper that needs to solve the proximal operator for a novel model). However, given the fundamental importance of the potential applications (e.g., Gaussian likelihood estimation), I would imagine this work would be of interest to the community even given the limited scope. Minor points/typos: a) In the equations after lines 71 and 74: prox_\phi (X, Y) ---> prox_\phi (X, y) b) Adding a comment that the final equality in the equation above line 81 comes from (12) and the fact that \mu^* is a root of (12) would be beneficial to the reader. c) C(\mu) is used in Theorem 4, but C(\mu) is not defined until below Theorem 4 (lines 159-160).

Correctness: I read through the main result (Theorem 1) is some detail and am convinced it is correct. I did not notice any errors in the remaining results, but I also did not read them in as careful detail.

Clarity: With the exception of the introduction I thought the material was well presented and the proofs were clear and simple to follow. I have left suggestions for how the introduction could be improved above.

Relation to Prior Work: The relationship to prior work was well explained.

Reproducibility: Yes

Additional Feedback:


Review 3

Summary and Contributions: This delightful paper uses classic ideas in convex analysis to discover a new characterization of the proximal operator matrix perspective function as a univariate minimization problem. Using this characterization, they develop an efficient Newton method to compute the prox, and show how to use it to solve several interesting machine learning problems.

Strengths: The paper is a delight to read, and presents the theory very clearly. The experiments are also well thought out. The topic is somewhat classical, but important and should be useful to the NeurIPS community.

Weaknesses: A few small aspects of the presentation could be improved. * line 48: clarify in which reference this iteration appears * line 53: "reassures" is not the right word. * display above line 81: I didn't quite follow this derivation. * explain more the innovation of this work compared to the previous works [25] and [3], and perhaps others. (In particular, the paper overall has no related work section, though relevant papers are cited along the way.) * line 169: Convex.jl deserves a citation, just as you do for MOSEK.

Correctness: Yes, the paper seems complete and correct.

Clarity: Yes, a delight to read.

Relation to Prior Work: This could be improved. Explain more the innovation of this work compared to the previous works [25] and [3], and perhaps others. (In particular, the paper overall has no related work section, though relevant papers are cited along the way.)

Reproducibility: Yes

Additional Feedback:


Review 4

Summary and Contributions: In this work, the proximity operator of the matrix perspective function is analyzed. This proximity operator is show to have link to well-known learning tasks, such as lasso and graphical lasso. Next, it is shown that computing the proximity operator of this function leads to solving a root finding problem, and when found the operator takes a closed-form solution. The proximity operator is connected to the matrix nearness problem. Based on this connection, a Guard Newton algorithm is proposed to find the root (that converges quadratically by further analysis). Last, the method is empirically evaluated.

Strengths: - The proximal operator analysis and its theoretical grounding - Contributions seem novel

Weaknesses: - Empirical evaluation is limited to only one baseline and synthetic data - No empirical analysis on the convergence - Discussion about the meaning of the theoretical results would be helpful to understand the steps. - It is hard to tell the relevance of the specific proximity operator as all problems have alternative solutions. Those solutions were not included in the empirical evaluation.

Correctness: The claims and method seem valid to the best of my knowledge and understanding.

Clarity: It was very difficult to understand the paper and details are dense. The Introduction can be presented in a more organized manner, and separated in well defined paragraphs. The background details can be presented as such.

Relation to Prior Work: The prior work is included in the paper with regard the theory seems fine. The additional optimization problems and their empirical evaluation could use a better set of references. (see below for some)

Reproducibility: No

Additional Feedback: UPDATE: I have read the authors' responses and the other reviews. Thanks for clarifying my confusion between the problems. I have updated my score accordingly. ---------------------------- For example, FISTA utilizes the proximal operator defined directly from the objective function. Therefore, why would someone choose to modify the problem to fit the proposed proximal operator instead of the other existing methods? The authors compare to MOSEK, which is an SDP solver. SDP solver are well known to require high amounts of memory. On the other hand, many algorithms exists that can solve these problems in much higher dimensions and with very acceptable runtimes. Some examples for the graphical lasso include [1], [2], [3]. FISTA and Non-linear Conjugate Gradients [4]. How do these methods compare to the Guarded Newton-based proximal method? It would be nice to see experiments with real-world data. Also, convergence plots comparing methods would be useful. What are the specs on the laptop? Memory? [1] Sparse Inverse Covariance Matrix Estimation Using Quadratic Approximation, Hsieh et al., 2011 [2] BIG & QUIC: Sparse Inverse Covariance Estimation for a Million Variables, Hsieh et al., 2013 [3] A Block-Coordinate Descent Approach for Large-scale Sparse Inverse Covariance Estimation, Treister and Turek, 2014 [4] L1-L2 Optimization in Signal and Image Processing, Zibulevsky and Elad, 2010

[Author Response · NeurIPS 2020]

1 We thank the reviewers for their thoughtful comments. To summarize, there is a consensus that the proximal analysis
2 is theoretically sound and novel and that our method outperforms existing ones. Most concerns are regarding the
3 presentation, scope, and technicality, not about the content. We believe concerns can be largely addressed by better
4 exposition and targeted clarifications.

**Presentation**: We believe this submission is the first to notice the importance of the prox operator of the matrix
perspective function $\phi$. Refs [2, 19] only mention that $\phi$ is convex and stop there. We got interested in the prox of
this function in need for constrained joint MLE of the two natural parameters of multivariate Gaussian (Eq 2). Most
textbooks (e.g., [And09]) deal with fairly simple settings that can be solved analytically, or submit to suboptimal
procedures. Proximal methods come to rescue for more complex settings at modern scales. Yet, computing the prox of
$\phi$ turned out to be nontrivial, let alone doing it efficiently and accurately. We will add this point to §1 and reorganize it
after merging §2 and §3, as suggested by R1. (Minor typos have already been fixed.)

**Scope**: Given the significance of the multivariate Gaussian, we think enlarging the class of solvable estimation problems
is important and useful to the community, let alone other problems discussed in the submission. Joint estimation of
Gaussian natural parameters under constraints have not received much attention, and we think a part of the reason is the
lack of practical optimization algorithms. Please see Comments 3-3 by R2 and 2-2 by R3.

**Technicality**: We understand the concern on the high technicality of the submission. On the other hand, we think it is
the nature of semismooth optimization. As the latter is starting to get applied to learning problems [LST18, CZST20],
we also think this submission is timely to the community. We will add remarks on the meaning of the technical results
as recommended by R1 and R4. They are roughly as follows. *Theorem 1*: computation of the prox operator of the
nonsmooth function $\phi$, which does not have a closed form, reduces to a univariate root-finding problem. *Theorem 2*: the
function whose root is sought is nonsmooth, but strongly semismooth and satisfies conditions for a Newton algorithm to
exist and converge locally. *Theorem 3*: convergence is global and the rate is asymptotically quadratic. *Theorem 4*: the
subgradients that need to be computed for the semismooth Newton algorithm have a closed form.

**R1**: **1)** We will merge §2 and §3 to avoid redundancy. **2)** Proof of Thm 3: Yes, $0 \leq v_k \leq 1$ for all $k$. It follows from the
Bolzano-Weierstrauss Theorem that $\{v_k\}$ has a limit point. Presentation will be improved once §2 and §3 are merged
to give more space. **3)** Convergence result in Thm 3 is *global*, while asymptotic. The nonasymptotic analysis in [2]
relies on strong convexity and Lipschitz continuity of the Hessian, neither of which does not apply to our problem. In
many semismooth Newton literature [25, LST18, CZST20], the analysis is either local or asymptotic. (Asymptotically)
quadratic convergence usually translates to that only a few iterations are required to get high accuracy, as can be seen in
Table 1. Any (including stochastic) proximal algorithm will benefit from a fast, high-accuracy prox routine, since it is
the former that is to run for a finite but large number of iterations.

**R2**: We think the motivation for the Newton method is delineated in the above "Presentation" paragraph. Bisection was
ruled out since a) in a similar matrix nearness problem [3, §4] it was reported to be orders of magnitudes slower than
Newton (when the latter is possible), and b) when we observed that the Newton algorithm took less than 10 iterations for
all experiments with several orders of magnitudes better accuracy (in terms of the KKT measure) than the interior-point
solver MOSEK, we thought that the game was over. We will add bisection to Table 1 and potentially convergence plot
similar to [3, Fig 3] as suggested by R4.

**R3**: Comparison with [3] and [25] – Our dual (Eq 15) is similar to the "one rank-one" case of [3]. However, in [3, §3.4],
that $\bar{\mathbf{X}} \succeq \mathbf{0}$ plays an important role in devising a (smooth) Newton algorithm; we do not assume this. Furthermore,
quadratic convergence of the Newton algorithm is only claimed and not proved in [3]. In [25], the constraint is that all
diagonal entries of $\mathbf{U}$ are 1. Rather surprisingly, our constraint that only one diagonal entry is 1 makes the perturbation
analysis of the spectral decomposition of $\mathbf{C}(\mu)$ more difficult (Lemma A.3 and §A.4) than [25, Lemma 3.4].

**R4**: **1)** None of the three learning tasks illustrated in the submission possesses an obvious alternative solution method
like FISTA or non-linear conjugate gradient acceleration of ISTA. ISTA-type methods require the objective to be the sum
of a smooth and nonsmooth functions, where the former has a Lipschitzian gradient. Foremost, function $\phi$ is nonsmooth.
Thus the *heteroskedastic* scaled lasso (Eq 4), unlike the plain lasso, cannot benefit from ISTA. In this case, a popular
method is Chambolle-Pock [5], a special case of PDHG (see L47-48 and references therein). Variance-constrained
joint Gaussian MLE (Eq 2) and pseudolikelihood graphical model selection (Eq 3) are *not* the same as the graphical
lasso. The differentiable parts of the objectives do not have Lipschitzian gradients due to the logarithmic terms, hence
PDHG was considered. Even in the graphical lasso, if the location parameter ($\boldsymbol{\eta}$) is to be jointly estimated, then the
log-likelihood (+ $\ell_1$ penalty on $\boldsymbol{\Omega}$) still contains the $\phi$. **2)** For the convergence plot, see our response to R2.

[And09] Anderson, T. W. An Introduction to Multivariate Statistical Analysis. Wiley, 2009.
[CZST20] Chu, D. et al., "Semismooth Newton Algorithm for Efficient Projections onto $\ell_{1,\infty}$-norm Ball." ICML 2020.
[LST18] Li, X., Sun, D. and Toh, K.C. "A highly efficient semismooth Newton augmented Lagrangian method for
solving Lasso problems." SIAM Journal on Optimization, 28(1), pp.433-458, 2018.


[Meta-Review · NeurIPS 2020]

R1 thought the paper was good but perhaps too narrow in scope, though R1 was positively moved by the rebuttal. R2 agreed that motivation was the biggest weakness and otherwise was very positive. R3, agreeing on the technical points, thought there was enough applicability to NeurIPS. R4 significantly raised their score after the rebuttal clarified some issues, and agreed with R3 that there was enough applicability. Overall, these were nice results, and the reviewers were quite positive.